# The Memory of Hops: Rural Bioculture as a Collective Means of Reimagining the Future

Estrella Alfaro-Saiz [1,*], Susana Cámara-Leret [2], Miguel González-González [3], Óscar Fernández-Álvarez [3], Sergio Rodríguez-Fernández [1], Darío López-López [1], Ana I. Paniagua-García [4], Carmen Acedo [1] and Rebeca Díez-Antolínez [4]

1    Department of Biodiversity and Environmental Management, University of León, 24071 León, Spain; srodf@unileon.es (S.R.-F.); dlopel01@estudiantes.unileon.es (D.L.-L.); c.acedo@unileon.es (C.A.)
2    Independent Researcher, 24153 Vegas del Condado, Spain; info@susanacamaraleret.com
3    Social Anthropology, University of León, 24071 León, Spain; migog@unileon.es (M.G.-G.); oscar.fernandez@unileon.es (Ó.F.-Á.)
4    Centre of Biofuels and Bioproducts, Agricultural Technological Institute of Castilla y León, Villarejo de Órbigo, 24358 León, Spain; pangaran@itacyl.es (A.I.P.-G.); dieantre@itacyl.es (R.D.-A.)
*    Correspondence: estrella.alfaro@unileon.es

**Abstract:** This article, set within an art–science collaborative framework, exposes a multidisciplinary research platform aimed at identifying new relationships with hops (*Humulus lupulus*), its harvest, and local memory. It presents an ecological and ethnobotanical study of the plant, from its natural habitat to its past/present cultivation, its traditional uses, and possible applications in pharmacy and cosmetics. It offers a qualitative study with an ethnographic approach to participant observation, using techniques such as in-depth interviews, focus group discussions, and life stories. Finally, it brings forth a process of material experimentation from the arts, based on rethinking waste systems to create new biomaterials with manifold future applications. The results from this hybrid methodology show the multiple possibilities that the plant offers beyond its cultivation for the brewing industry. Likewise, it shows how spaces, relationships, and dialogues have been generated with wide repercussions on a local and planetary scale, related to the sustainability of the rural world and territorial cohesion, all of which are intrinsic to emergent agrarian practices. The conclusions show a complex scenario that demands a hybrid response to understand the paradoxes to which the plant is subjected and the uncertain future of agrarian culture.

**Keywords:** rural knowledge; thinking-through-making; loss and damage; sustainability; rural livelihoods; agroecology; herbarium; living collection; memory bank; agricultural waste

## 1. Introduction

"Green gold" is the term locally used to denote the cultivation of hops (*Humulus lupulus* L.) in the province of León (Castilla y León, Spain) due to its socioeconomic relevance. This term originated in the 1980s when the crop was harvested across 1950 hectares in Spain, primarily in the León region [1]. This species, deeply rooted in this area, is notably prevalent along its principal riverbanks—the Tuerto, Órbigo, Torío, Porma Bernesga, and Cea rivers—and has modified and conditioned the lives of its inhabitants (Figure 1).

The following two forces drove the expansion of hops in Spain: the free choice of farmers to adopt this crop and the production needs of the brewing industry [1]. Following World War II, the "Sociedad de Fomento del Lúpulo" was established, later succeeded by the "Sociedad Anónima Española de Fomento del Lúpulo" (SAEFL) in 1946, headquartered in Villanueva de Carrizo, León. This organisation received a national monopoly for hop cultivation and processing. Hop cultivation in León was encouraged by research on local flora along the Bernesga, Torío, and Órbigo rivers [1,2]. Early trials in A Coruña

between 1904 and 1928 [2] laid the groundwork for introducing 'Hallertau', 'Golding', and 'Fino Alsacia' hop varieties from Europe in the 1940s. The 1960s development of 'Northern Brewer' and 'Brewer's Gold', later known as 'H-3' and 'H-7', alongside 'Tettnang', became León's leading strains [3]. From the 1960s onwards, the need to create bitter hop varieties intensified due to the demands of the brewing industry, albeit preserving their aromatic properties. This led to the introduction of hop varieties such as 'Nugget', currently the most cultivated variety in León, 'Magnum', 'Columbus', 'Eureka', 'Chinook', and 'Golden Brewer'.

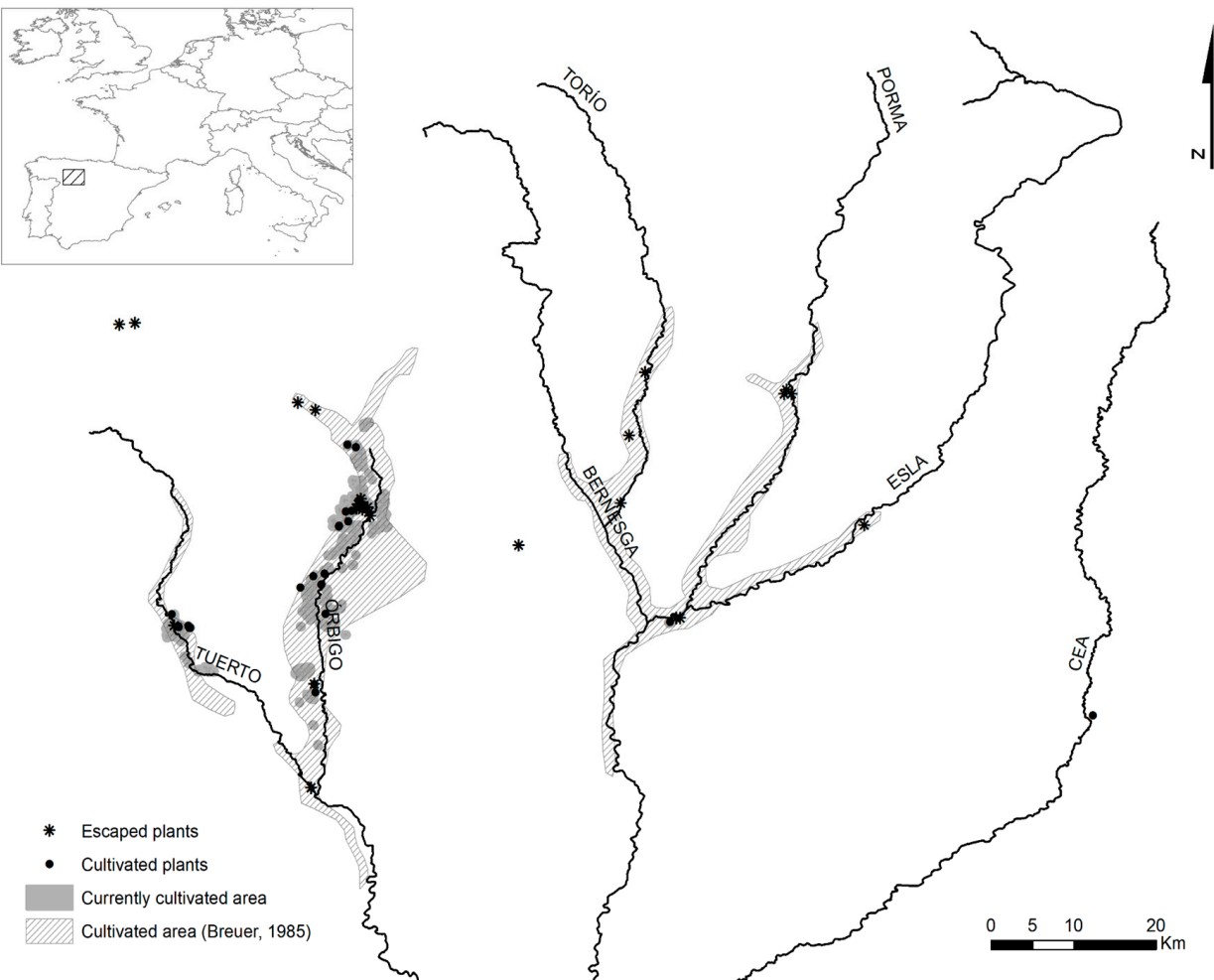

**Figure 1.** Map showing the past and present harvest of hops throughout the main riverbanks of León [1].

Most hop production is dedicated to the brewing industry, which continues to influence the varieties grown, their cultivation, and market prices. In 2016, the multinational Hopsteiner acquired 80% of SAEFL's shares, with the remaining 20% belonging to the local Agrarian Transformation Society (Sociedad Agraria Transformadora, SAT) Lúpulos de León. Presently, this multinational's presence, spanning over 500 hectares, accounts for 97% of Spain's hop production [4]. Spain ranks as the ninth largest global hop producer and the second in Europe, following Germany [5]. Despite the significance of León's hop production for the Spanish agri-food sector and its economic importance to local residents, there has been little to no significant promotion or modernisation of its production, harvesting, and drying processes. While certain phytosanitary products and drying methods have been updated to less harmful alternatives, these changes were primarily in response to legislation rather than efforts to enhance the production system.

Rural areas like León cover more than 80% of the Spanish territory, serving as key producers of essential resources and food while embodying a rich, intangible biocultural heritage. However, their socioeconomic structure is fragile. The phenomenon known as "emptied Spain" features an ageing population and local agricultural systems—a sector often viewed as strategically important—struggle under market forces and capitalist economies that demand continued economic growth within a globalised food system [6]. The ongoing departure of younger generations from rural areas exacerbates a scenario marked by declining populations due to rural–urban migration, posing further risks to biocultural diversity [6] and the sustainability of rural livelihoods. This complex scenario requires a hybrid response to understand the paradoxes threatening local, place-based biocultures alongside the plant's agricultural landscapes and systems. This study presents an art–science collaboration from the "Cultivate Cultures: Ecologies of Hops" project, which, over two years, has established a multidisciplinary platform for research, experimentation, and participation to explore sustainable relationships with the plant's harvest and image.

### 1.1. Conceptual Framework

Knowledge of nature accumulates within a society and is often shared through narratives, storytelling, and observations [7–9]. It forms a cultural understanding of nature that shapes landscapes through individual and collective actions [10,11]. Landscapes, created by life processes, serve as records, testimonies, or archives of past generations [10,12]. A landscape is a story, and perceiving it is "to carry out an act of remembrance" [12].

Escaped hop plants populate many of León's riverbanks. Known as naturalised plants, these species are introduced into ecosystems different from their original ones, either intentionally or accidentally, for ornamental or agricultural purposes. They can spread uncontrollably, competing with native species, impacting biodiversity and ecosystems negatively [13], interfering with ecological processes and the genetics of native species, and becoming hosts for diseases or pests. These plants can rapidly colonise new territories due to their reproductive, dispersal, and adaptation abilities. While some ornamental species can spread vigorously with significant ecological and economic effects [14], in degraded or currently cultivated environments, or areas of abandoned agricultural activity where the species is not invasive and is controlled, escaped *Humulus lupulus* plants in León are considered a local small-scale resource.

Cultural and biological diversity are closely connected [15–17], with a broad recognition that the diversity of life is fundamental to the diversity of cultural practices and worldviews. Recent international policies, such as UNEP's (2007) Global Environmental Outlook (GEO), have advocated for the protection of both biological and cultural diversity, describing biodiversity as essential to "human cultural diversity, which can be affected by the same drivers as biodiversity, and which has impacts on the diversity of genes, other species, and ecosystems" [18]. Despite this, cultural diversity is often seen as less critical than biodiversity protection [6], prompting calls for a "paradigm shift to consider biological and cultural diversity as one" [6]. Preserving biocultural diversity hinges on sustainable livelihoods, defined as "environmentally sustainable when it maintains or enhances the local and global assets on which livelihoods depend, and has net beneficial effects on other livelihoods" and "socially sustainable when it can cope with and recover from stress and shocks, and provide for future generations" [19]. The external pressures of monopolised, unsustainable agricultural production, low wages, and rural depopulation, as seen in León's hop harvesting communities, threaten their sustainability and traditional cultures, risking the homogenisation of local rural culture and landscapes.

Linked to key elements such as the preservation of biocultural diversity and sustainability, Silva, Biase, and Martellini [20] discuss agroecology as an approach integrating ecological, social, and economic principles to promote sustainability [21] and equity in food systems. This requires the interaction between organisms and resources—such as soil, water, and the natural environment—to prevent biodiversity loss or social inequality in food production and access, as highlighted by Altieri and Toledo [22]. This approach values

traditional practices and knowledge of rural communities and emphasises the need for collaboration and dialogue among various actors, including farmers, scientists, consumers, cultural practitioners, and policymakers, as proposed in this paper. UNESCO defines intangible cultural heritage as transmitted from generation to generation, constantly recreated by communities and groups in response to their environment, interaction with nature, and history, fostering a sense of identity and continuity while promoting respect for cultural diversity and human creativity [23].

Moreover, this discussion includes materiality, referring to the quality or characteristic related to the physical or tangible aspects of objects, structures, or elements [24,25]. The recent shift towards new materialism [26] considers the effects of "naturecultures", a term by Haraway [27] aiming to transcend the nature/culture dichotomy. In our case study aligned with agroecology, bioculture, and rural knowledge, materiality is closely related to how elements like seeds, hop plants, soil, tools, and production methods influence agricultural practices and community relationships with plants and their environment, aiming for sustainable resource use and minimising environmental impacts [25].

Additionally, rural knowledge, linked to popular culture and ethnographic concepts like ethnobotany, traditional agriculture, and vernacular languages, addresses people's relationships with their natural environment and traditions, safeguarding their tools, practices, and traditional uses. This traditional knowledge, at risk of being forgotten, is crucial for sustainability and accessing endogenous resources from natural sources for society. By integrating agroecology, bioculture, and rural knowledge, we promote more holistic and collaborative approaches to managing agricultural systems and conserving the rural environment. This involves valuing rural knowledge, encouraging community participation, and linking local with scientific knowledge to tackle current agricultural and sustainability challenges [28]. In this study, we equate rural knowledge with citizen science, recognising rural inhabitants as key custodians and information providers for science, knowledge, and society.

In summary, it is essential to consider this theoretical framework, including materiality, agroecology, bioculture, and rural knowledge, as study references. The use, management, and relation of materials to the natural environment and cultural practices significantly affect the sustainability, resilience, and preservation of agricultural systems and rural communities.

*1.2. Objectives*

The search for sustainable, future-based alternatives grounded in nature enables us to explore pressing socio-cultural issues associated with the image of hops. These are still considered an identity symbol along the Órbigo riverbank in León, where production is now primarily focused, yet they also evoke strong memories along the other riverbanks where they were once harvested.

To investigate these aspects, we propose a multidisciplinary study that utilises various methodologies to describe the human–plant relationships central to hop harvesting in León:

(1) The creation of a biocultural archive on the plant's memory from a multidisciplinary perspective, including herbarium vouchers of cultivated, wild and escaped plants, samples for molecular studies, and a living collection of cultivars.

(2) An ethnobotany of *Humulus lupulus*, its habitat, and the past and present of its cultivation, including the study of its traditional uses and possible applications in pharmacy and cosmetics. We address existing knowledge of the plant's chemical composition, active ingredients, and properties, with a beneficial nature for humans, validate or refute the therapeutic uses of phytochemicals through a literature review, and confirm the prediction with in silico simulation of molecular targets. In addition, we study the composition of some crop varieties in the province of León and also escaped specimens.

(3)  A qualitative study from an ethnographic approach based on participant observation, following the criteria established by Hammersley and Atkinson [29], where techniques such as in-depth interviews, focus group discussions, and life stories are used.

(4)  An art-led research and experimentation process, aimed at readdressing the crop's waste through a "hands-on" approach and the creation of biomaterials, artefacts, and narratives, to reimagine existing relationships with the plant aligned to the sustainability of its biocultures.

## 2. Materials and Methods

### 2.1. Collection of Hop Samples

a.   Herbarium specimens

As part of the material recorded, herbarium voucher specimens of different hop samples (*Humulus lupulus* L.) were collected, pressed, and prepared, including both current and traditional varieties. The latter were preserved by farmers among their crops, with some specimens being over 50 years old, noteworthy considering the maximum harvest peak is 20 years, after which production declines. Additionally, samples from wild escaped plants likely originating from old plantations and now part of the riverbanks were collected. Two duplicates of each specimen were made, one for the project and another for the institutional collection of the Herbarium LEB Jaime Andrés Rodríguez. Ultra-dried material was also collected for future molecular studies to analyse genetic variability and enrich the memory bank.

b.   Material was also sent by volunteers through a citizen science campaign, equipped with materials and data collection instructions. The campaign was promoted via social networks, herbaria networks, and scientific societies such as AHIM—Asociación de Herbarios Ibero Macaronésicos.Living collection of cultivars.

Living collections of different hop varieties were also collected as part of the material recorded for the project. Thanks to the collaboration of some farmers, it was possible to find some of the traditional crop varieties that were grown at the facilities of ITACyL and the Faculty of Agricultural Engineering of the University of León.

c.   Samples for extraction

Samples of 500 g of fresh female inflorescences were collected from different cultivated and escaped varieties from the province of León for the drying, extraction, and analysis of their bioactive components. The inflorescences were taken to the Villarejo de Órbigo Biofuels and Bioproducts Research Center (CBB) for dehydration at 45 °C and subsequent analysis (Point 2).

d.   Collection of the harvest's waste for material experiments (stems and leaves)

Whilst hop flowers are highly priced by the brewing industry, the vegetative parts of the plant (the stems and leaves) are considered valueless. During harvest, the peelers separate the female inflorescences from the stems and leaves that accumulate to form considerable piles of wasted matter. Local farmers generally leave it to ferment and dry, later burying it in nearby lands or burning it, together with the plastic cords used to climb the plant, locally known as "climbers" or "trepas". The "residues" were collected, both whole leaves and stems, and in some locations, already shredded.

### 2.2. The Analysis of Bioactive Compounds

The extraction of essential oil was performed by ITACyL at the Biofuels and Bioproducts Research Center (CBB) in Villarejo de Órbigo (León), using steam distillation with a Clevenger apparatus according to European Brewery Convention (EBC) standard 7.10 [30]. To identify concentrations of $\alpha$ and $\beta$ acids, the essential oil's composition, and xanthohumol, high-performance liquid chromatographic analysis (HPLC) was utilised, following EBC methodologies 7.7, 7.12, and 7.15, respectively («Analytica EBC ∣ Hops and Hop Products ∣ 7.7—$\alpha$- and $\beta$-Acids in Hops and Hop Products by HPLC», ref. [31];

«Analytica EBC | Hops and Hop Products | 7.12—Hop Essential Oils by Capillary Gas Chromatography Flame Ionisation Detection», ref. [32]; «Analytica EBC | Hops and Hop Products | 7.15—Xanthohumol in Hops and Hop Products by HPLC», ref. [33]). Gas chromatography was used to determine the molecules' concentrations in the samples, diluting the essential oil in an organic solvent and using a flame ionisation detector. Results were expressed as a percentage of the dry weight of the collected samples.

In addition to the compounds analysed by ITACyL, others cited in the literature were included for their high concentration in the plant and potential health benefits as follows: 2-methyl-3-buten-2-ol, 8-prenylnaringenin, germacrene-D, humulol, alpha-pinene, lupulone, and isoxanthohumol.

The Swiss Target Prediction (STP) tool [34], employing an in silico validation methodology based on the "principle of similarity" between molecules, was used to predict the likelihood of two bioactive molecules sharing macromolecular targets (proteins or targets). This prediction relies on a combination of 2D and 3D similarity, with a library of 370,000 known substances in over 3000 proteins from three species: human (*Homo sapiens*), mouse (*Mus musculus*), and rat (*Rattus norvegicus*) [35].

The PubChem molecular database was used to search for each selected molecule to collect their SMILE code, representing the chemical structure with alphanumeric strings.

### 2.3. Ethnographic Interviews

In-depth interviews were carried out, following the criteria established by Hammersley and Atkinson [29], with eight active farmers and four knowledgeable experts in the history of hop cultivation in the area, mostly men due to the characteristics of the agricultural activity and the social context of the rural environment. Likewise, they also participated in debates and discussion groups in which, through an open script (*filandón*), the information provided flowed as the conversation progressed. In this sense, it was possible to investigate various relationships with the plant, paying special attention to issues such as motivation, feelings, acculturation, or economic, social, and cultural growth, on a personal, family, or community level.

Representativeness was sought, in access to the population under study, which was manifested from the understanding of the relevant social relations refs. [35,36] analysed in the previous ethnographic perspective, where a structural heterogeneity in the object of study was perceived. Subsequently, the saturation point was established [37], which determined the sample size. In this way, a solid basis is provided for the legitimation of qualitative research, even validating the emotions of the informants.

There was informed consent, where all people agreed to participate voluntarily in this research and knew the criteria and purposes of the project.

Audiovisual material was generated, for which the work team collaborated with Ismael Aveleira for filming and Rafael Martinez del Pozo for sound recording.

### 2.4. Art-Led Material Experiments

A series of material experiments were performed with the plant matter that is currently leftover during hop harvest. The aim was to explore the creation of new biomaterials from what is currently considered waste, as a means to amplify the plant's imaginaries. The work was performed through an art-led explorative process, where making and thinking go hand-in-hand: a "thinking-through-making" approach. This process included the development of collective gatherings and workshops with local residents and farmers, opening up the material experimentation process to the public whilst stimulating discussions on the current cultivation system. For the purpose of this study, we focused on the following material experiments and activities: the extraction of fibre from stems; the creation of paper pulp from stems; textile dyeing with stems and leaves; weaving with stems; audiovisual and sound productions; and the combination and application of the resulting biomaterials in workshops held within the *Antruejo* celebratory activities, a winter carnival period and local rural festivity held in the town of Carrizo de la Ribera, on the Órbigo riverbank.

a.    Extraction of the plant's fibre for textile production

There is historical evidence of the use of hop fibres for the development of textiles [38]. Hops' structure is similar to bast fibre plants like hemp or linen, with an outer bark and an inner pith [39]. Although there are few examples, previous experiences show that the fibrous outer bark can be used to produce long-length fibres (10–15 cm) with properties similar to that of hemp [39].

A similar procedure was followed for the processing of hemp fibre. Leftover plant matter was collected after the harvest. The branches and leaves were separated, and the stems were left to dry. Once dried, they were soaked or retted in a nearby irrigating canal. After 2–4 weeks, plant material was dried for several weeks and stored in a dry area prior to the breaking, or scutching, of the decomposed stalks to separate the fibres from the woody matter.

b.    Textile dyes from the leaves and stems

Initial experiments were performed by extracting colours from the dried stems, fresh and dried leaves, and fresh male inflorescences. Dried female inflorescences were also tested as a reference, following local farmers' observations on the flower's stains. For the experiments, we collaborated with art educator Ana Andrés Cristobal.

Different fabrics were used for dyeing including cellulose fibres, such as a 50–50% linen/cotton blend; and protein-based fibres, including 100% merino wool, Bourette silk (made from silk waste), and felted wool samples extracted from an old mattress. To prepare the fabrics for dyeing, various combinations of mordants were used, including Alum (20%), a combination of Alum (20%) + Cream of tartar (8%) (colour changer), Oak Galls (20%), Tannic Acid (20%), and Vinegar and Salt (25%).

All fabrics were firstly de-gummed. For this, we used a mixture of 5 L of water, one teaspoon of soap, and 3 teaspoons of Solvay soda per 100 g of fabric and let it soak for 24 h. The cellulose fibres were additionally given a protein bath of soymilk, with a mixture of 5 L of water and ½ L of soymilk per 100 g of fabric, in which they were left to soak for 24 h, removed, dried, and left to rest for 7 days. For the oak galls, we boiled 20 g oak galls per 100 g of fabric, boiled in 1 L water. After boiling it for 1 h, we covered the batch with more water, let it rest for 24 h, removed the fabric, and let it dry.

To prepare the fabric for dyeing, we used 800 mL of distilled water per fabric sample. The following mordants were used: Alum (20%), a combination of Alum (20%) + Cream of tartar (8%) (colour changer), Oak Galls (20%, pH = 8), Tannic Acid (20% pH = 8), Vinegar (until covered, pH = 3) and Salt (25%, pH = 7). We allowed the mixture to boil, let it cool, and then submerged the fabric for one hour.

During the dyeing process, the plant matter was cut up and left to soak in water for 24 h. We used the same amount (g) of dried plant matter to the weight of the fabric, and in the case of fresh plant matter, we used twice the amount. It was heated just under boiling temperature. After 1 h, the fabrics were introduced into the bath and dyed for 30 min.

c.    Paper pulp

A similar procedure to paper-making techniques from plant fibres was used, working with the plants' stems [40]. Firstly, the plant stems were dried, cut into 3 cm. pieces, and left to soak in water for 24 h. They were then boiled for 4 h (1 kg of plant matter, 15 L water) and left to rest in the same water for 12 h. The softened stems were then blended to obtain a paper pulp. No additional materials were used to extract the lignin. Wooden frames were built with a thin wire mesh, which was used to extract the paper leaves from the pulp and water mixture.

Larger panels, of a greater thickness, were also produced as a first attempt to explore its potential application as an ecological bioconstruction isolating material. For this purpose, two 300 × 300 × 30 mm samples were produced and tested with the Instituto Eduardo Torroja—CSIC (*Centro Superior de Investigaciones científicas*) to obtain the thermal conductivity coefficient according to the UNE-EN 12667:2002 standard [41] (Determination

of thermal resistance by the flow metre method) for a material sample, for products of high and medium thermal resistance.

d.    Weaving with the plants' stems

To weave with the plants' stems, a similar process to wicker was followed. For these experiments, we collaborated with basket weaver Carlos Fontales, who has been researching traditional rural weaving techniques for over 30 years. The hops stems were firstly dried and, prior to weaving, soaked in water for 24–48 h.

e.    Audiovisual production

Throughout the realisation of the interviews, fieldwork, material experiments, and workshops, film and sound recordings were performed to document the activities and narratives of local farmers and residents. The filming was performed by Ismael Aveleira and the sound recordings by Rafael Martínez del Pozo.

f.    *Antruejo*

Three workshops were organised in collaboration with the Cultural Association La Trepa from Carrizo de la Ribera, in the context of the rural winter carnival festivities known as the *Antruejo*. This is a rural tradition full of iconography and symbolism intimately linked to agrarian culture and its cycles, which happens over several days just before Lent. Amongst the different activities, there is a parade in which the traditional characters of the *Antruejo* parade throughout the town, accompanied by music.

The first workshop focused on making paper masks from hop pulp and had an average of 10 participants. The second workshop centred on dyeing larger fabrics to create garments, working with the plant's stems, leaves, and other residues from the harvest, in collaboration with art educator Ana Andrés Cristóbal, and had an average of 34 participants, mainly consisting of local neighbours and farmers. Finally, a third workshop was performed together with basket weaver Carlos Fontales. We worked with two basic techniques of circular motion including the following: the coil technique and the chickpea knot, with a total of 45 participants, mainly consisting of local neighbours and farmers.

**3. Results**

*3.1. Collection of Hop Samples*

a.    Herbarium specimens

A total of 102 herbarium specimens were collected (Appendix B and Figure 2), each with its duplicate. In total, 32 specimens are of 17 cultivated varieties, 30 are of plants cultivated on farms, and two are cultivated as ornamentals. The remaining 70 specimens corresponded to plants growing in the natural environment, representing escaped plants whose origin is uncertain but probably derived from old plantations.

During field surveys, some male plants were located and collected. Although male plants are physically removed from hop fields to avoid fertilisation of female plants and thus seed production, male plants are essential in hop breeding programs to develop new varieties through controlled hybridisation [42], and their presence could be indicative of the presence of wild individuals.

b.    Living collection of cultivars

The following traditional varieties were obtained and cultivated for their preservation in a scientific collection: 'H-7', 'H-3', 'Eureka', 'Chinook', 'Nugget', 'Apolo', and 'Tettnang', as well as some varieties escaped from riverbanks.

c.    Samples for extraction

Twenty-eight samples of inflorescences were collected. Eight of these derived from plants that grew outside plantations, and the remaining 20 belonged to crop varieties from different producers.

d.    Collection of the harvest's waste for material experiments (stems and leaves)

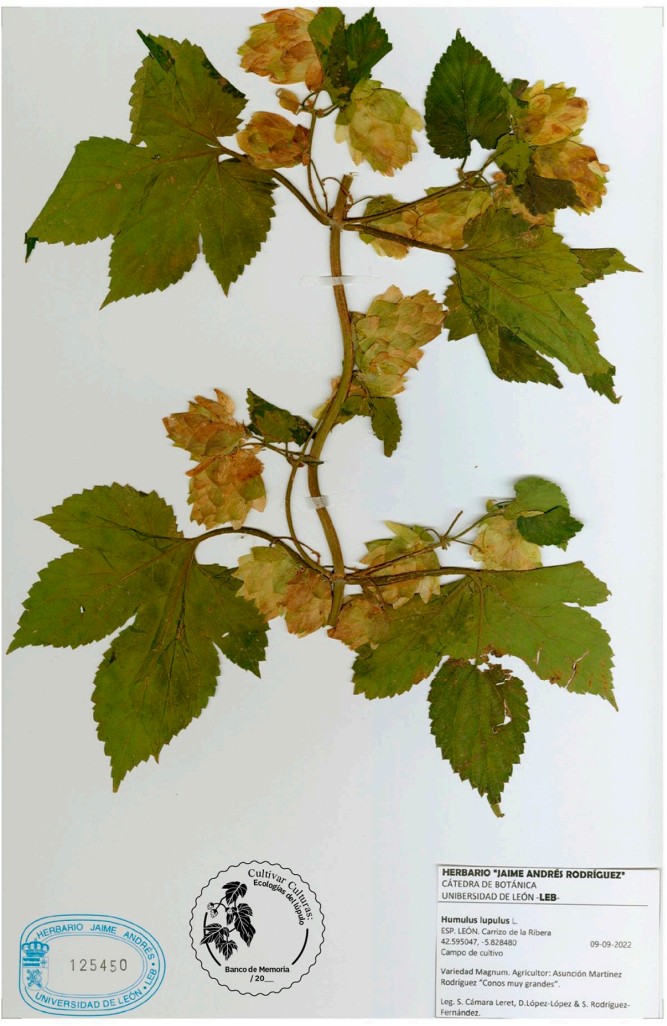

**Figure 2.** A herbarium voucher of a 'Magnum' specimen collected from Carrizo de la Ribera, including habitat information and the farmer's observations regarding this variety's: crop field (campo de cultivo) and large cone sizes ("conos muy grandes").

These samples were used to carry out the material experiments based on readdressing the harvest's waste (see Section 3.4).

### 3.2. Analysis of Bioactive Compounds

The twenty-two main compounds present in the essential oil of *Humulus lupulus* and the biological activity of these molecules, which could explain the therapeutic uses reported for hops, are shown in Appendix B. In addition to showing the chemical groups and chemical classes to which the studied compounds belong, the anti-inflammatory, anticancer, antimicrobial, antioxidant, and sedative activities proven by clinical trials are highlighted.

The results obtained using CBE methodologies are shown in Table 1. Twenty-eight samples of female inflorescences were analysed. The levels of essential oils, α- and β-acids, and xanthohumol were also measured. The biomolecules found within the analysis were studied, along with others found in the literature.

**Table 1.** Summary of the samples analysed: essential oil content, essential oil composition, α- and β-acids and xanthohumol. LEB is the acronym for herbarium Jaime Andrés Rodríguez of the University of León. WV means a sample without a voucher. nd means a sample without determined data. Ls means lack of sample.

| Herbarium Number | Variety | Essential Oil Content | Essential Oil Composition | | | | | | | | | A-Acids Data | B-Acids | Xantho-humol |
|---|---|---|---|---|---|---|---|---|---|---|---|---|---|---|
| | | | β-Pinene | Myrcene | Limonene | Linalool | Geraniol | 2-Undecanone | β-Caryophyllene | β-Farnesene | Humulene | | | |
| LEB 124964 | Alsacia | 1.82 | 0.47 | 35.74 | 0.33 | 0.26 | 0.19 | 0.21 | 9.63 | nd | 19.64 | 10.51 | 4.39 | 0.72 |
| LEB 124998 | H 3 | 1.66 | 0.74 | 55.58 | 0.48 | 0.33 | 0.24 | 0.05 | 8.60 | 0.02 | 19.80 | 5.36 | 4.55 | 0.79 |
| LEB 125028 | H 7 | 2.21 | 0.72 | 53.95 | 0.46 | 0.32 | 0.02 | 0.50 | 8.15 | 0.02 | 21.79 | 9.13 | 4.45 | 0.91 |
| LEB 125014 | Escaped plant | 0.28 | 0.05 | 1.86 | nd | 0.08 | 0.24 | 1.44 | 6.92 | 20.75 | 5.13 | 2.21 | 5.00 | 0.17 |
| LEB 124909 | Escaped plant | 0.72 | 0.19 | 15.12 | 0.13 | 0.24 | 0.12 | 0.54 | 15.47 | 5.92 | 38.74 | 5.68 | 3.67 | 0.56 |
| LEB 125016 | Columbus | 1.21 | 0.15 | 15.00 | 0.13 | 0.39 | 0.06 | 0.33 | 16.06 | nd | 31.51 | 11.42 | 4.36 | 0.90 |
| LEB 125015 | Nugget | 1.81 | 0.49 | 41.72 | 0.31 | 0.83 | 0.02 | 0.45 | 11.89 | 0.02 | 24.51 | 10.53 | 3.97 | 0.81 |
| LEB 124967 | Nugget | 1.85 | 0.55 | 48.34 | 0.36 | 0.76 | 0.02 | 0.43 | 11.03 | 0.02 | 22.20 | 11.00 | 3.91 | 0.79 |
| LEB 124968 | Nugget | 1.10 | 0.19 | 19.76 | 0.16 | 0.5 | nd | 0.52 | 17.40 | nd | 34.12 | 9.82 | 4.34 | 0.85 |
| LEB 125020 | Nugget | 1.49 | 0.20 | 22.71 | 0.20 | 0.74 | nd | 0.61 | 16.52 | 0.03 | 32.25 | 11.95 | 4.26 | 0.85 |
| LEB 125019 | Escaped plant | ls | 0.03 | 2.72 | 0.02 | 0.14 | 0.30 | 1.24 | 9.83 | 11.63 | 17.77 | 3.14 | 4.26 | 0.39 |
| LEB 124963 | Pacific Gem | ls | The profile could not be determined due to lack of sample | | | | | | | | | 5.27 | 6.30 | 0.63 |
| LEB 125021 | Chinook | 2.19 | 0.59 | 41.23 | 0.37 | 0.44 | 0.38 | 0.23 | 9.04 | nd | 19.04 | 10.16 | 3.82 | 0.72 |
| WV | Nugget | 2.37 | 0.59 | 52.06 | 0.37 | 0.88 | 0.03 | 0.37 | 9.13 | nd | 18.60 | 11.03 | 3.88 | 0.83 |
| LEB 124961 | Aquila | 1.45 | 0.69 | 54.73 | 0.49 | 0.53 | 0.33 | 0.52 | 7.92 | nd | 17.12 | 7.34 | 3.77 | 0.66 |
| LEB 124961 | Aquila | 1.36 | 0.73 | 59.28 | 0.51 | 0.47 | 0.41 | 0.96 | 6.91 | nd | 13.72 | 7.95 | 3.73 | 0.64 |
| LEB 125022 | Millenium | 1.28 | 0.53 | 39.29 | 0.32 | 0.67 | 0.11 | 0.40 | 10.99 | 0.02 | 24.61 | 9.05 | 3.71 | 0.67 |
| LEB 125022 | Millenium | 1.63 | 0.42 | 35.22 | 0.31 | 0.30 | 0.11 | 0.27 | 10.87 | 0.02 | 24.28 | 10.15 | 4.72 | 0.72 |
| WN | Golden Brewer | 1.11 | 0.56 | 39.21 | 0.36 | 0.33 | 0.14 | 0.34 | 8.40 | 6.73 | 15.55 | 4.10 | 3.95 | 0.42 |
| WN | Golden Brewer | 0.77 | 0.21 | 18.42 | 0.20 | 0.27 | 0.08 | 0.45 | 10.99 | 8.88 | 22.40 | 3.51 | 3.67 | 0.39 |
| LEB 125034 | Escaped plant | 0.91 | 0.29 | 27.33 | 0.17 | 0.43 | nd | 0.95 | 11.37 | 0.19 | 20.30 | 3.63 | 6.24 | 0.35 |
| LEB 125033 | Escaped plant | 0.64 | 0.20 | 21.65 | 0.13 | 0.33 | nd | 1.29 | 7.69 | 13.37 | 13.01 | 4.70 | 5.29 | 0.42 |
| LEB 125035 | Escaped plant | 0.74 | 0.49 | 46.62 | 0.30 | 0.75 | nd | 1.22 | 3.85 | 16.11 | 8.57 | 4.43 | 4.45 | 0.26 |
| WN | Escaped plant | 0.89 | 0.40 | 37.54 | 0.24 | 0.38 | 0.02 | 0.52 | 3.66 | nd | 17.10 | 3.72 | 4.85 | 0.32 |
| LEB 125003 | Eureka | 3.15 | 0.60 | 55.69 | 0.40 | 0.57 | 0.02 | 0.27 | 9.00 | 0.12 | 17.94 | 18.75 | 5.20 | 0.65 |
| LEB 125003 | Eureka | 1.87 | 0.50 | 46.89 | 0.34 | 0.74 | nd | 0.23 | 9.16 | 0.50 | 19.30 | 12.67 | 4.56 | 0.85 |
| LEB 125027 | Eureka | 3.30 | 0.61 | 53.95 | 0.40 | 0.59 | 0.03 | 0.24 | 9.11 | nd | 20.47 | 18.03 | 4.99 | 0.89 |
| LEB 125007 | Escaped plant | ls | The profile could not be determined due to lack of sample | | | | | | | | | 3.58 | 5.09 | 0.34 |

Only the targets with the highest probability of binding with the molecule analysed for *Homo sapiens* species were chosen to analyse their pharmacological uses. Molecular targets with a probability of less than 50% were discarded, so biological properties cannot be assumed to have action. The results (see Table 2) show that the main targets of the 8-prenylnaringenin molecule are oestrogen receptors alpha and beta (96.71%), ATP-binding cassette transporters, Subfamily G, Member 5 (90.03%), and Cytochrome P450 (74.14%). Some of the twenty-two molecules of *Humulus lupulus*, cohumulone, adhumulone and β-farnesene did not give results, so the prediction was not performed. For the molecules α-caryophyllene, D-germacrene, humulol, lupulone, α-pinene, β-pinene, myrcene, limonene, geraniol, linalool, 2-undecanone, adlupulone, colupulone, and xanthohumol, the probability was less than 50%. Bindings with 0.0% probability were disregarded, as in the case of the molecule 2-methyl-3-buten-2-ol.

**Table 2.** Targets with the highest probability of binding to those of the molecule analysed for the *Homo sapiens* species.

| Molecule | Target | Common Name | CHEMBL ID | Target Class | Probability |
|---|---|---|---|---|---|
| 8-prenilnaringenina | Oestrogen receptor alpha | ESR1 | 206 | Nuclear receptor | 0.9671 |
| 8-prenilnaringenina | Oestrogen receptor beta | ESR2 | 242 | Nuclear receptor | 0.9671 |
| 8-prenilnaringenina | ATP-binding cassette sub-family G member 2 | ABCG2 | 5393 | Primary active transporter | 0.9003 |
| 8-prenilnaringenina | Cytochrome P450 19A1 | CYP19A1 | 1978 | Cytochrome P450 | 0.7414 |
| β-cariofileno | Peroxisome proliferatoractivated receptor alpha | PPARA | 239 | Nuclear receptor | 0.7192 |
| β-cariofileno | Cannabinoid receptor 2 | CNR2 | 253 | Family A G protein-coupled receptor | 0.7192 |
| kaempferol | NADPH oxidase 4 | NOX4 | 1250375 | Enzyme | 1.0 |
| kaempferol | Aldose reductase (by homology) | AKR1B1 | 1900 | Enzyme | 1.0 |
| kaempferol | Xanthine dehydrogenase | XDH | 1929 | Oxidoreductase | 1.0 |
| kaempferol | Tyrosinase | TYR | 1973 | Oxidoreductase | 1.0 |
| kaempferol | Tyrosine-protein kinase receptor FLT3 | FLT3 | 1974 | Kinase | 1.0 |
| isoxanthohumol | Cytochrome P450 19A1 | CYP19A1 | 1978 | Cytochrome P450 | 0.7018 |
| isoxanthohumol | Oestrogen receptor alpha | ESR1 | 206 | Nuclear receptor | 0.6612 |
| isoxanthohumol | Oestrogen receptor beta | ESR2 | 242 | Nuclear receptor | 0.6612 |

Two binding targets obtained for β-caryophyllene have probability percentages of 71%. The main targets of the β-caryophyllene molecule are the alpha receptors activated by peroxisome proliferators (71.92%) and type II cannabinoid receptors (71.92%).

In the case of kaempferol, one hundred binding targets were obtained, with probabilities ranging from 100% to 17%. The first seventeen results have a binding probability of 100%. Shown in the table are the first five of the sixteen results with 100% probability. The targets of the molecule are multiprotein complexes responsible for producing reactive oxygen species, aldose reductase, xanthine dehydrogenase enzymes, tyrosinase, and receptor tyrosine-protein kinase FLT3.

In the case of isoxanthohumol, ninety-six binding targets were obtained, with probabilities between 70% and 10%. The first four results have a probability of union with their objectives of between 70% and 43%. The first three targets obtained for the isoxanthohumol molecule are cytochrome P450 (70.18%) and oestrogen receptors alpha and beta (66.12%).

*3.3. Ethnographic Interviews*

Although the plant was considered "green gold" at a time when it was the only livelihood for some families and, for others, a complement to their economic activity, the results from the fieldwork show that its social and economic considerations indicate a

simultaneous love and hate relationship with the plant. In a favourable domain, the plant serves as an economic activity due to its good economic performance shortly after the start of the harvest and mainly during its seasonal dedication. This positive affective component was also shown when the informants talked about their family trajectories within the recent history of ho harvest in their area since there are several families and producers who claim to be pioneers in the crop's cultivation in their region. Likewise, these feelings were also evoked in what is supposed to be a traditional activity and its seasonality. For example, when speaking about the festive celebrations during the harvest and the manual peeling of hop flowers.

Negative evocations refer to a labour point of view due to the work involved and the care the plant requires, in such a way that, although cultivation began in the 1950s, no more than three generations have been involved in its cultivation. Along the same lines, family conflicts arise due to socio-labour aspects or the generational takeover, which are linked to the decision-making capacity of producers regarding their production and commercialisation options. These tensions are due to modernisation, changing conditions in the production system, and the fluctuating value of product yields depending on production costs.

As a direct outcome of the ethnographic interviews, a thematic profile was constructed, correlating closely with both the narratives of the reporting agents and the research criteria applied. (Table 3).

**Table 3.** Patterns and actors with whom the topics were mostly discussed.

| Patterns | Actors | Topics |
|---|---|---|
| Descriptive | - Local governance.<br>- Cultural association.<br>- Retired farmers.<br>- Individual farmers.<br>- New generations of farmers. | - Social and economic considerations with the plant.<br>- Relationship with the plant.<br>- Considerations on the cultivation and traditional knowledge of hops.<br>- Festive celebrations. |
| Theoretical | - Family context.<br>- Retired farmers. | - History of hop harvest in their area.<br>- Considerations on depopulation and sustainability. |
| Political | - Local governance.<br>- Cultural associations.<br>- Hop farmers associations (S.A.T.).<br>- New generations of farmers. | - Tensions and conflicts due to modernisation.<br>- Fluctuating value of products.<br>- The relations established with the multinational Hopsteiner.<br>- Considerations on depopulation and sustainability. |
| Personal | - Family context.<br>- Individual farmers.<br>- Hired workers. | - Family trajectories (generational succession).<br>- Festive celebrations.<br>- Considerations on depopulation and sustainability. |
| External | - Experts. | - Considerations on depopulation and sustainability. |

Going into further detail on this matter, we observe the following three fundamental aspects: the considerations on the cultivation and traditional knowledge of hops, the relations established with the multinational Hopsteiner, and the considerations on depopulation and sustainability.

The traditional cultivation and artisanal hops present notable differences compared with modern cultivation. Both processes comprise key stages, such as site selection, soil

preparation, rhizome planting, plant care and maintenance, training for proper branching, and harvest, which usually takes place in late summer or early fall when the cones reach maturity.

The traditional cultivation of hops in León, and specifically in Carrizo de la Ribera, implemented a manual and artisan approach, where farmers invested time and effort to guarantee the quality of the hops. Although modern techniques have introduced technological advances, many still value and remember the tradition and craftsmanship of hop production. The testimonies of the hop harvesters highlight manual peeling as an especially arduous task in the traditional cultivation of hops, although they all agreed that the toughest work performed during that time was "digging it", that is, removing the head of the plant.

> "Hasta que vino el tractor o la maquinaria todo se hacía a mano, con la azada, también con la guadaña"

> ("Until the tractor or the machinery came, everything was done by hand, with the hoe, also with the scythe").

> "Antes apañábamos las piedras, de las tierras, ahora viene un tractor y las apaña".

> ("Before we removed the stones from the land, now a tractor comes and removes them.").

Here, we observe typical Leonese speech, where the meaning of pick up is said as "*apañar*" instead of "*recoger*". In this sense, it is possible to consider the traditional cultivation of hops as part of the intangible ethnographic heritage, referring to practices, expressions, knowledge, and skills transmitted from generation to generation within a community, which are recognised as an integral part of their cultural identity.

In relation to the traditional cultivation of hops, some testimonies from the hop harvesters recall songs sung during the manual peeling of the flowers, which highlights the cultural dimension and the importance of traditional practices in collective memory. It is also necessary to mention that there is an idealisation or romanticism of the past when manual cultivation was physically more demanding, although it also fostered social relations and solidarity amongst workers. Many "lupuleros" and "lupuleras" (male and female hop harvesters) remember that time with certain nostalgia.

> "Si no hubiera sido el trabajo de todas estas mujeres y estos hombres que han trabajado antes el lúpulo, los que trabajamos ahora el lúpulo no estaríamos aquí"

> ("If it hadn't been for the work of all those women and men who worked hops previously, those of us who work hops now wouldn't be here")

A significant testimony of a *lupulera*, obtained in the *filandón* carried out in Gavilanes, recalls the songs sung during the manual peeling of hops, which tells us about the social conditioning imposed by agricultural work, especially in periods of greater activity.

> "Como quieres niña que te vaya a ver,

> Si vengo del campo al anochecer.

> Cuando llego ceno y arreglo el ganado,

> Cuando voy a verte ya te has acostado.

> Ya te has acostado no quieres abrirme,

> No sabes las penas que paso por ti."

> "How do you want me to come see you, girl?

> If I come from the field at dusk.

> When I arrive I have dinner and groom the cattle,

> When I come to see you, you've already gone to bed.

You've already gone to bed, you don't want to open up,

You don't know the sorrows I go through for you."

In addition to the personal changes, as indicated by the previous example, there is also abundant information obtained that speaks of social changes. For example, it is pointed out that there is currently a greater individualism in the cultivation of hops, in contrast to the value that was given previously to collectivity and service to the community. This can be seen in the decline in collaborative practices, such as "hacenderas", historical rural traditions in the region of León. (*Hacenderas* are collective workdays in which the inhabitants of a rural community come together to help each other in agricultural and livestock work or in making improvements to shared infrastructures.) Moreover, it can also lead to personal conflicts among producers or tensions caused by the multinational Hopsteiner, as we will see hereafter.

In summary, the traditional cultivation of hops in León presents specific characteristics that can be considered as part of the intangible ethnographic heritage. The second element of consideration is the existing relations with the multinational Hopsteiner. During the fieldwork, it was verified that the role of the multinational buyer of local hops is fundamental, whilst at the same time generating great controversy and conflicts among producers. These types of companies play a prominent role in the supply chain and marketing of hops at a national and international level. They are usually key players in the brewing industry and are in charge of acquiring and processing the hops grown by local producers. Some of the most relevant aspects to consider are as follows: Firstly, the purchasing and commercialisation of hops since this multinational directly acquires the product from the producers, establishing commercial agreements and purchase contracts. This provides stability and market guarantees, as well as security in their operations. In addition, they have access to the global supply chain and can transfer knowledge and technology to local producers.

However, this role of the multinational as a buyer of local hops has also generated debates and discrepancies amongst hop harvesters. Tensions have mainly been generated due to the pressure to maintain competitive prices and meet the quality and sustainability requirements demanded by these companies. For example, a recurring theme is the discussion on the use of sustainable materials to climb the plant, such as natural fibres instead of plastic, or the production of organic hops.

It is important to keep in mind that although multinational buyers of local hops do influence pricing, they are not the only ones responsible for determining it. Other factors such as supply and demand, production costs, hop quality, and market conditions also have an impact. In this respect, the testimony of a *lupulero* during the *filandón* de Carrizo de la Ribera is interesting, as he stated: "La multinacional marca los tiempos y las relaciones, incluso diría que hasta las disputas que hay en el pueblo". ("The multinational marks the times and the relationships, I would even say that also the disputes that exist in the town.")

The third element for these considerations is depopulation and the significant impact on the social and cultural structure of the rural communities on the Órbigo riverbanks and, in general, of the rural environment. The decrease in population has resulted in accelerated demographic ageing, with a reduced presence of young people and a lack of generational relief. This has generated implications for community life, including a decline in traditional social and cultural activities, as well as in the vitality and cohesion of communities. A shared concern about depopulation has been observed amongst hop harvesting areas, a common phenomenon in other rural areas as we have pointed out.

Some testimonies collected reflect this concern: "*Y que no se pierda (el cultivo del lúpulo), ¿verdad? Que no se pierda. Porque es una pena que los pueblos queden vacíos y al menos así, los jóvenes se quedan. Si hay matrimonios jóvenes, hay niños, hay vida. Los mayores ya, por mucho que...*" ("And that it isn't lost (hops cultivation), right? It should not be lost. Because it's a pity that the towns remain empty and at least that way, the young people will stay. If there are young married couples, there are children, there is life. The older ones already, no matter how much...")

### 3.4. Art-Led Material Experiments

a.     Extraction of the plant fibre for textile production.

Fine fibres were extracted from the plant stems, of 10–15 cm in length, which were suitable for spinning and weaving. These were of a warm beige colour, similar to linen. Shorter felted fibres were also obtained, similar to scutching tow in the processing of flax fibre (see Figure 3b).

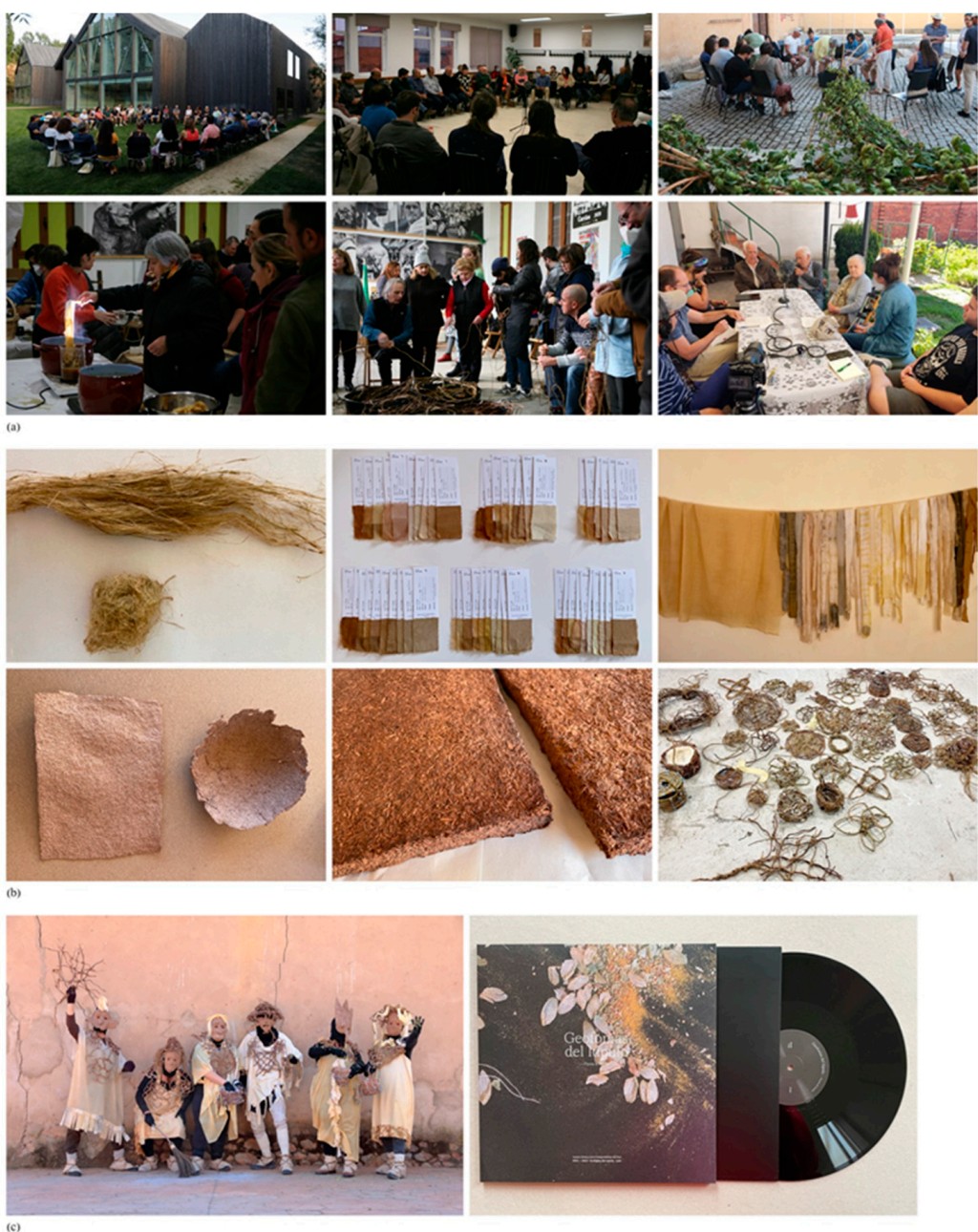

**Figure 3.** The art-led experiments involved various processes and outcomes: (**a**) Public gatherings, interviews and workshops; (**b**) experiments with the plant's wasted matter to obtain fibre and colour dyes and exploring its use for weaving; and (**c**) the "Motas and Manchas", characters created from hop harvest residues in the context of the rural winter carnival *Antruejo*, and the "Geophonies of Hops: Flowers and Sleep" sound publication addressing new aesthetic relationships with the plant aimed at amplifying its iconography and imaginaries.

b.   Textile dyes from the leaves and stems

Colour represents the presence of different plant bio-dye molecules. In total, 107 different colour samples were obtained, ranging from browns to pinks and yellows (see Figure 3b). Using a qualitative analysis, we observe that hop stems produce a tan-brownish colour, whilst the leaves produce a yellow colour similar to the one obtained from dried female flowers. These colour tests are suitable for textile dyeing.

c.   Paper pulp

A series of paper samples were obtained of different thicknesses. During the process, boards of a more flexible consistency were created, blending hop stems with coconut rope used by young farmers interested in ecological harvests. Boards of a stiffer and thicker consistency were also produced, similar to that of cardboard and packaging materials, along with paper objects (plates and vases) (see Figure 3b). From 1 kg of plant matter, two $300 \times 300 \times 30$ mm tiles and two $200 \times 100 \times 70$ mm bricks were produced with a considerable amount of leftover material for the creation of additional objects and paper sheets. The preliminary results of the tests performed on two thicker ($30 \times 30$) tiles show that the average thermal conductivity coefficient of the hops stem paste sample is W/mK 0.0531.

d.   Weaving with the plant stems

Several spiral-like motifs were collectively created during the workshop. These were later combined to create different accessories for the six Antruejo characters, each with their hop-dyed gown and hop paper pulp mask (see Figure 3b,c). Participants also created woven baskets and hats, experimenting with the stems, and combining these with the plastic ropes used to climb the plant.

e.   Audiovisual and sound production

Five audiovisual short clips were made, registering the harvest of the plant and its life cycles. These are available on the project's YouTube channel (@ecologiasdellupulo4443) and include:

La poda (abril 2022, 3′12″) [Pruning];

El Entrepado (abril 2022, 1′31″) [Climbing the plant];

La cosecha (agosto 2022, 2′56″) [The harvest];

Entre tiempos (abril 2023, 1′59″) [In-betweens];

La cuerda (abril 2023, 1′30″) [The rope].

From all the recordings, a documentary was also produced titled *No Sólo Flores (Not only flowers)*, which reflects on the tensions observed between the plant's harvest and the transformation of local rural, agrarian culture. It includes testimonies from farmers of an advanced age, linked to the origins of the harvest in León, as well as younger generations currently active and facing its future challenges. It also depicts the activities and processes developed with local communities during this research, aimed at rethinking the plant's imaginaries.

Next to this, an ambient, sound publication was produced titled "Geophonies of Hops: Flowers and Sleep" (Geofonías del Lúpulo: Flores y Sueño) (see Figure 3c). It consists of a double vinyl (45 rpm, LP 12″). Side A contains a composition produced from field recordings. Side B includes a sound composition made from sounds produced from the weight of the hop flowers.

f.   Antruejo

The materials (hop paper masks, dyed gowns, and woven structures) resulting from this research and the three public workshops were used to create six characters for the *Antruejo* festivities at Carrizo de la Ribera (see Figure 3c). They took part in the winter carnival parade and were nicknamed the "Motas and Manchas", as *mota* is the popular

term used for flower and *mancha* means to stain, given that the hop flower is known to stain the clothes and hands of those who harvest it.

## 4. Discussion

The results highlight concerns on a European scale regarding rural sustainability and territorial cohesion [43], recognised as international issues by the European Union. These challenges can be addressed through elements identified in our findings, such as rural entrepreneurship, co-creative practices, and the management and utilisation of endogenous resources [44,45]. The social economy is a crucial factor for rural development [46], with entities dedicated to its advancement capable of generating stable employment and revitalising the territory. This involves promoting productive diversification and social cohesion. Effective rural development necessitates the collaboration of various economic, social, and political agents to maintain productive capacities and enhance developmental opportunities for individuals [47] while integrating innovation into sustainable development strategies and processes.

This transdisciplinary approach highlights the importance of preserving traditional agricultural practices while exploring sustainable and innovative uses of agricultural by-products. Emphasising the need for a balanced integration of cultural values, economic viability, and environmental sustainability, this discussion advocates for collaborative efforts across disciplines to enhance the resilience and sustainability of rural communities. It underscores the potential of traditional crops like hops to contribute to contemporary challenges, including biodiversity conservation, cultural preservation, and the development of bio-based materials, thus paving the way for a more sustainable and culturally enriched future.

### 4.1. Biocultural Approaches

In this respect, from an anthropological perspective, bioculture reveals how the biological and cultural entanglements of hops mutually influence individuals' lives, becoming integral to their rural memory. The division between nature and culture is artificial, as human experience is deeply embedded in an uninterrupted existence in the world. This existence is shaped by belief systems, social practices, history, symbolic imagination, technologies, language, and the general cultural environment. Such divisions can be transcended through practice and learning [48].

Biocultural approaches to studying hops encompass various aspects that align them with sustainability science, creating a conceptual and practical framework for integrating different academic disciplines and non-academic perspectives, as this study aims to accomplish. Furthermore, as Hanspach, Haider et al. [49] suggest, biocultural approaches should co-produce knowledge for sustainability solutions, addressing issues related to gender, power, and transformative action still underrepresented in mainstream discussions.

The transdisciplinary approach discussed here suggests novel participation frameworks, where negotiation processes are vital. These negotiations encompass formal aspects, such as identifying frameworks and defining protocols and sequences of actions. Engaging in negotiations with public administrations, farmers, foundations, anthropologists, biologists, artists, neighbourhood associations, and other community members requires integrating diverse languages and sensitivities.

Kremen, Iles, and Bacon [50] highlight the importance of agroecology in linking diversified, transformed, and reoriented agricultural systems. These systems aim for new uses, utilisation, and reuse of hop waste and residues to prevent stagnation and become part of socio-ecological systems that rely on specific knowledge, culture, practices, management structures, and local and traditional governance of contemporaneity, thereby enhancing sustainability.

The diversification and reorientation of hop agriculture relate to preserving traditional crop varieties or exploring new opportunities arising from the abandonment of certain agricultural practices where "escaped" plants flourish. Exploring new alternatives for

hops involves studying the plant's cultural heritage and cultivation history. Cultivated hop varieties offer potential applications in antimicrobial, cancer, and metabolic syndrome treatments, and hormone replacement therapies and are used in insecticides, preservatives, fragrances, or foodstuffs. Future hop breeding efforts should leverage genetic resources, including wild populations and local varieties [51].

*4.2. Biocompounds and Cultivars*

The herbarium specimens collected during our fieldwork are the vouchers of the localities we confirmed. In addition, these specimens allowed us to link the current presence of the species with the crop map published by Breuer in 1985 for the province of León [1]. The absence of samples in areas where crops did not exist leads us to consider the possibility that the plant is no longer in the wild.

Clinical trials reveal that *Humulus lupulus* phytochemicals like 8-prenylnaringenin, kaempferol, β-caryophyllene, and isoxanthohumol have substantial health benefits, including anti-cancer, anti-inflammatory, and antioxidant effects, significantly influencing the nervous, digestive, and endocrine systems [52–61]. Hops, used for treating insomnia, menopause, and other conditions [62,63], involve key compounds like isoxanthohumol and β-caryophyllene, with the former, a xanthohumol isomer, acting as a potent agent against menstrual pain and potentially in cancer prevention by affecting oestrogen metabolism [64,65]. Analysis of 'H7', 'Columbus', and 'Nugget' hops revealed high xanthohumol levels, fully converting to isoxanthohumol upon boiling, which confirms and enhances its bioavailability. 'Nugget', 'Columbus', and wild León hops had the highest β-caryophyllene levels. Farmers reported drowsiness during harvest and in hop kilns, likely due to 2-methyl-3-buten-2-ol, a compound increasing γ-aminobutyric neurotransmitter (GABA) production, thus sedating and promoting sleep by altering circadian rhythms [58]. Other compounds, including α- and β-acids, act as sedatives, antidepressants, hypnotics, analgesics, anxiolytics, or neuroprotectors by inhibiting the CNS's cholinergic system and possessing antioxidant properties [52,59].

Residents used hop flowers to enhance digestion and stimulate digestive juice secretion, attributed to bitter compounds like α- and β-acids and β-caryophyllene [66]. Hop extracts, demonstrating carminative and spasmolytic effects, inhibited acetylcholine production, relaxed muscles, and increased gastric secretions, effectively acting on gastrointestinal motility [67]. These extracts also showed antioxidant, cytoprotective, and anti-inflammatory benefits across the digestive system, except in the liver [68].

For the endocrine system, the use of *Humulus lupulus* is emerging, especially for women, as it contains potent phytoestrogen 8-prenylnaringenin, validated by STP results, and xanthohumol with lower estrogenic activities.

Besides healthcare, *Humulus lupulus* finds applications in cosmetics or perfumery, containing myrcene, linalool, and limonene. These molecules, not validated in STP simulations, can be applied topically, offering citrus and floral scents. High concentrations of these molecules are found in 'H3', 'H7', 'Nugget', 'Aquilla' and 'Eureka'.

Varieties like 'H3' and 'H7', now disused, were significant in León's agricultural landscape, alongside the highly cultivated 'Nugget'. Local varieties contribute to cultural identity and preserve traditional agricultural knowledge. Their conservation is crucial for genetic diversity, enhancing adaptability to different production systems and resilience against climate changes, thereby securing food security [69]. With rising temperatures and more frequent droughts, breeding efforts for hop cultivars suited to warmer climates are essential. Eriksen et al. [70] identified promising cultivars for such conditions.

Addressing the underuse of hop compounds in healthcare despite known benefits, this study highlights the pharmacological potential of *Humulus lupulus*, especially its bioactive compounds with therapeutic effects validated by clinical trials. Regulatory hurdles and the pharmaceutical industry's preference for synthetic drugs may restrict these compounds' use in mainstream healthcare. However, our findings recommend further exploration of hops as alternatives to harness these beneficial compounds.

### 4.3. Bioproducts vs. Waste

This study's findings indicate that hop by-products, typically considered waste, can offer versatile applications in several sectors, such as bio-dye molecules. These agricultural waste materials present bio-based solutions for the textile industry, mitigating water contamination risks from synthetic dyes in wastewater due to their high flavonoid and tannin content [71–73]. Flavonoids, particularly kaempferol in hop leaves and flowers, produce a yellow dye [72,74] and are recognised as a key chromophore in dyeing plants [75].

Furthermore, the plant's so-called waste materials, like the stems, have not always been considered so, as evidenced by its historical uses for textile production in Scandinavia, between the 15th and 19th centuries [38]. In this study, we show that the fibres obtained from the stem are suitable for spinning and weaving. As for the shorter fibres obtained, although they are not suitable for spinning, other applications could be recovered by revisiting the plant's historical textile uses. For example, hop stems and flowers were used as pillow and blanket padding in Danish farms until the 19th century [76]. Revisiting these practices, we propose that the shorter fibres obtained could also be used as a filler and/or padding material for various objects, to create felt-like surfaces, or even as composites for the stabilisation of papers and boards. Furthermore, as we have presented, hop stems can also serve as a natural biodegradable material for basket weaving to create artefacts and structures with potential architectural applications, for example, in the form of decorative panels, blinds, etc. Moreover, our results show that the shredded stems produced by some industrial peelers can be reutilised to create paper pulp of varying thicknesses for multiple applications, including the potential development of thermal/acoustic isolating materials for bioconstruction. As shown, hop paste presents a thermal conductivity of W/mk 0.053, similar, and possibly less, than other natural insulating materials such as expanded clay (W/mK 0.148–0.095) or cellular glass panels (W/mK 0.05). This warrants further research into the material's optimisation and standardisation to establish a definite comparison, something which we will continue to explore in the next phases of this project.

### 4.4. Brand Image

Harnessing artisanal hop cultivation as a resource for tourism can generate economic benefits, promote the preservation of cultural heritage, and enrich visitors' experiences by allowing them to immerse themselves in local agricultural culture. This approach can cover various facets, such as the creation of visitor centres and the promotion of rural tourism and its ethnographic valorisation. From an ethnographic perspective, artisanal hop cultivation and its traditional practices can be valued as cultural heritage and as an expression of local identities. This involves the recognition and preservation of the knowledge, techniques, and stories of hop cultivation passed down from generation to generation.

Hop cultivation has had a positive impact on the local economy by boosting job creation in the agricultural sector and promoting craft beer-related tourism, although to a lesser extent than might be expected as most of the hops produced in León are sold to multinationals. Some farmers offer guided tours of their hop fields, where visitors can learn about the growing process and the importance of hops in beer production. In the town of Carrizo de la Ribera, León, the Hops and Beer Fair has been annually held for over fifteen years, paying tribute to this characteristic product from the area and attracting a large number of visitors. Similarly, for example, in the town of El Bolsón, a centre of craft beer production in Argentina, a festival is held around hops and beer. These festivities allow us to learn about the cultural and production practices associated with the harvest of hops [77]. We can observe that such distant places have a similar relationship regarding the cultivation of hops and the beer festival. It is essential that these approaches are developed in a sustainable and respectful way, involving local communities whilst ensuring a balance between the preservation of local traditions and knowledge, and the adaptation to the demands of contemporary tourism.

There are numerous researchers who have addressed issues aligned with tourism's processes of differentiation and the transformation of places into globalised tourist attrac-

tions that configure a specific image of the world [78,79]. Some of the observed agricultural realities of hop cultivation in the province of León are bitter like the plant's organoleptic experience. This study's focus on the ecologies of hop's image considers ways in which the cultural productions of hops—its stories, experiences, and landscapes—are part of symbolic order and how they are assimilated within tourist attractions. The results visualise other uses and imaginaries of the plant beyond the beer industry, as a first step to perceive what lies beyond the permitted visibilities of its "brand image" [79] and promote more ethically responsible resources for tourism in the region. The Geophonies of Hops publication, for example, pays attention to the plant's aesthetic qualities and soundscapes. Sound is information but, as authors like Pauline Olivero reflect, habit has often "narrowed listening to only what seems of value and concern to the listener. All else is tuned out or discarded as garbage." [80]. Listening to hop's soundscapes offers a means to rethink the processes that tune out, or discard plant matter as waste, to consider more symbiotic relationships with the plant's harvest.

*4.5. Rural Memory and the Future of Hops*

Our results bring to the forefront the pressing need to preserve rural memory and its practices, to ensure a more sustainable use of local resources. Ecology means home, as authors like Lucy Lippard remind us, "a place that is difficult to find today" [81], linking the ecological crisis to the current concern for place and context, as well as the nostalgia caused by the loss of roots [81]. The plant's memory—its varieties, colours, fibres, smells, sounds, etc.—is proposed as a means to counteract the "loss and damage" to homes, livelihoods, ecosystems territory, languages and rural heritage sites caused by the extractive practices of the present harvest system. The "Motas and Manchas" series produced in this study seeks to expand the memory and iconicity of hops on the Órbigo riverbank, revisiting the local agrarian tradition of the *Antruejo*. The reuse of the crop's waste to create new artefacts activates other forms and aesthetics that amplify the plant's imaginaries.

These alternatives bring forth issues of governance, as discussions have also highlighted the existence of a possible colonial relationship between the multinational Hopsteiner and local producers. Hopsteiner controls the buying and selling of hops through long-term agreements, which make local harvesters highly dependent on its demands and decisions. It is significant how, in the *filandón* or discussion group held in Carrizo de la Ribera, most of the producers expressed the need for greater associationism and a more collaborative or cooperative culture. Although a local figure currently exists, the Agrarian Transformation Society Lúpulos de León, many consider that it is too closely aligned to the multinational and that small producers do not have enough influence. Blockades and confrontations have been identified in this regard, and some producers are afraid that hops will not be bought from them, which makes them yield to the demands of the multinational.

Other challenges and contradictions emerge with the lack of generational relief: although the inhabitants do not want the towns to lose population, they do not want their own children to continue agricultural tasks, preferring that they study in the city and not return to work in the fields. This lack of generational takeover mentioned above is contradictory to the complaints about the depopulation of towns. On the other hand, according to the information provided, they do not seem willing to pass on their knowledge and concede their lands to outsiders in order to avoid depopulation.

Despite the challenges posed by depopulation, it is important to note that resistant and resilient initiatives have also been observed by local communities. Some residents and groups have worked to revitalise hop regions, promoting economic diversification, the preservation of cultural heritage (such as the La Trepa cultural association), and the promotion of rural tourism. These initiatives seek not only to counteract depopulation but also to preserve the identity and sense of belonging of local communities. The recognition of these cultural practices and their appreciation contributes to the preservation of the rural identity and the historical legacy of the region, integrated into the rural knowledge of the community. The involvement of this association, the local producers themselves, and

the local community was essential in the engagement and development of workshops to explore new hop uses. Beyond its use in the brewing industry, its organic cultivation, or as a quality mark, it is here where the new possibilities offered by the plant are considered.

Cultural and agrarian traditions were once also radical innovations. The introduction of new technologies has been important to improve efficiency in hop cultivation. Some studies, including those carried out in other countries such as Italy, indicate that the use of new forms of mechanisation allows for a significant reduction in harvest time without damaging the cones [82]. In spite of this, there has been some resistance amongst small producers in León when it comes to making investments in technology, since some see these as an expense instead of an investment for the future: "It is very expensive, for such a small production it is not worth such an expense". A study conducted in one of the major hop-growing regions in the United States, the Yakima Valley, Washington, analysed the economic and social impacts of hop-growing on the local community, including employment, income, tourism, and challenges facing the industry. The study concluded that innovation in hop plantations increases producers' profits, improves their ability to adapt to the environment, and contributes to increasing sustainability. This finding is of vital importance to achieve the objectives of sustainable rural development and so, in its study, it is essential that this idea is also taken on board by policymakers [83].

The future of hops is thus understood as a shared responsibility that involves an attempt at a coexistence between bioculture and biodiversity as a means of understanding the sustainability of the crop. This study, centred on León's traditional hop cultivation, presents a detailed exploration of its unique contribution to agricultural sustainability within a specific locale. Aside from offering valuable localised insights, we recall that León produces 95% of all Spanish hop production and it is the tenth European producer. Therefore, this study's geographic specificity is also relevant to the applicability of our findings to other regions. Future research should aim to compare these practices across varied geographical contexts to extend the relevance of sustainable agricultural practices in other places.

## 5. Conclusions

*Humulus lupulus* is a species that presents clear advantages when designing and implementing new socio-ecological models in which knowledge, culture, agricultural practices, management structures, and governments come together in the search for real solutions in the face of an agri-food crisis. The possibilities of diversification, new applications, or production systems within a circular market for this species are very high. This is related to the multitude of uses it presents and to a not only feasible but recommended revaluation of the waste or by-products generated in its main industries.

Harvest residues contain a series of molecules that preserve odours, colours, flavours, and fabrics that serve as containers of rural knowledge. The consideration of the remains of the cone's harvest as waste speaks of a loss of agrarian culture and its ways of doing, which, as already demonstrated in this work, safeguards the many possibilities of the plant.

This approach bridges cultural heritage, sustainable agriculture, and innovative material uses, underscoring the interplay between traditional practices and modern sustainability challenges. By valuing diversity in all its forms—culture, languages, economic activities, cultivation systems, and crop varieties—it fosters a holistic understanding of hop cultivation's impact on its territory. This strategy not only promotes resilience and sustainability in rural communities but also serves as a model for establishing territorial guidelines and pilot models across different geographical contexts. This convergence of disciplines offers a path towards a sustainable, culturally rich future, highlighting the transformative potential of traditional crops like hops in addressing contemporary challenges.

**Author Contributions:** Conceptualisation, E.A.-S., S.C.-L., M.G.-G. and Ó.F.-Á.; data curation, E.A.-S., S.C.-L., M.G.-G., Ó.F.-Á., A.I.P.-G., C.A. and R.D.-A.; formal analysis, E.A.-S., S.C.-L., M.G.-G., Ó.F.-Á., A.I.P.-G., C.A. and R.D.-A.; funding acquisition, S.C.-L.; investigation, E.A.-S., S.C.-L., M.G.-G., Ó.F.-Á., S.R.-F., D.L.-L., A.I.P.-G., C.A. and R.D.-A.; methodology, E.A.-S., S.C.-L., M.G.-G.,

Ó.F.-Á., A.I.P.-G., C.A. and R.D.-A.; project lead and administration, S.C.-L.; resources, S.C.-L., E.A.-S., M.G.-G., Ó.F.-Á., A.I.P.-G., C.A. and R.D.-A.; software, E.A.-S., D.L.-L., A.I.P.-G., C.A. and R.D.-A.; supervision, E.A.-S., S.C.-L., M.G.-G., Ó.F.-Á., A.I.P.-G., C.A. and R.D.-A.; validation, E.A.-S., S.C.-L., M.G.-G., Ó.F.-Á., A.I.P.-G., C.A. and R.D.-A.; visualisation, E.A.-S., S.C.-L. and S.R.-F.; writing—original draft, E.A.-S., S.C.-L., M.G.-G., Ó.F.-Á., S.R.-F. and D.L.-L.; writing—review and editing, E.A.-S., S.C.-L., M.G.-G., Ó.F.-Á., S.R.-F., A.I.P.-G., C.A. and R.D.-A. All authors have read and agreed to the published version of the manuscript.

**Funding:** This research was developed within the art-led, multidisciplinary project *Cultivate Cultures: Ecologies of Hops*, supported and funded by Fondation Daniel et Nina Carasso, with the co-finance of the Diputación de León and the Ayuntamiento de Carrizo de la Ribera conceptualised and led by Susana Cámara Leret.

**Institutional Review Board Statement:** All procedures performed in studies involving human participants were in accordance with the ethical standards of the institutional and/or national research committee and with the 1964 Declaration of Helsinki and its later amendments or comparable ethical standards. This article does not contain any studies with animals performed by any of the authors.

**Informed Consent Statement:** Informed consent was obtained from all subjects involved in the study.

**Data Availability Statement:** Data are contained within the article.

**Acknowledgments:** The following people, connected to hop cultivation and the history of hops in León, have contributed to the development of this work through their time and generosity: From Carrizo de la Ribera: Alfonso Álvarez García, Anunciación Martínez Rodríguez, Bernardo García Peláez, Carlos García Muñiz, Daniel Fernández Rodríguez, Francisco de Paz García, Francisco Jimeno Peláez, José Fernández Fernández, and Sara Fernández Rodríguez. From La Milla del Río: Ángel Álvarez González, Alfredo Álvarez González, Desiderio Alonso González (R.I.P.), Gabriel González Llamas, Jessica Gordaliza Hidalgo, Jorge Pérez Martinez, Josefa González Blanco, Manuel González Alonso, Rodrigo Casas Fernández; From: Villanueva de Carrizo: Santiago Alonso González; From San Román de la Vega: Alberto Martínez Ferrero, and Sergio and Diego Sánchez García. From Benavides de Órbigo: César Fernández Viyeira, Javier Rubio Salgado, Javier Rubio García; From San Román de los Caballeros: Enrique Sevilla García, Yolanda Rubio Fuertes; From Cimanes del Tejar: Andrés García González, Eliodora Prieto Barreales, and Montserrat Álvarez Velasco. From Gavilanes de Órbigo: Isidoro Alonso Fernández, and Javier Fraile Marcos. From Secarejo: Pedro Díez Fuertes. From Llamas de la Ribera: Lourdes Gómez Román. From Cerezales del Condado: Jose Benjamín González González, Juan José González Díez, and María Ángeles González Robles. From Vegas del Condado: Rufino and Pablo Juárez and Esteban Otero Robles. From Ambasaguas de Curueño: Horacio Fernández Fernández and Nina Martínez Suárez. From Villanueva de las Manzanas: Gonzalo Pastrana García, Guillermo Marques Rodríguez, and Santiago Santos Valle. From Villoria de Órbigo: Manuel González Martínez. From Villamor de Órbigo: Juan Carlos Domínguez Matilla. From León: Eva López Campesino, as well as all the people who attended the *filandones* held in Carrizo de la Ribera, Gavilanes de Órbigo, Cerezales del Condado y Mansilla de las Mulas. Rafael Martínez del Pozo and Ismael Aveleira were an integral part of the work conducted between 2021 and 2023, based in Carrizo de la Ribera. We would also like to thank Borja Frutos Vázquez and Raquel Selfa Marugán from the Instituto Eduardo Torroja—CSIC. This work also received valuable support and collaboration from the following individuals and organisations: Ana Andrés Cristóbal, Carlos Fontales, Mónica Sisí, Asociación Cultural La Trepa de Carrizo de la Ribera, Fundación Cerezales Antonino y Cinia, and Museo de los Pueblos Leoneses. We extend our gratitude to Nieves Salgado Cubelos (Diputación de León) for her unwavering dedication and belief in this project.

**Conflicts of Interest:** The authors declare no conflicts of interest.

## Appendix A

Herbarium specimens collected during the fieldwork.

| Location | Province | Coordinates | Date (d/m/y) | Collectors | Observations | Voucher Number |
|---|---|---|---|---|---|---|
| Villanueva de las Manzanas | León | 42.4770 −5.4840 | 23/08/21 | E.Alfaro-Saiz, S.Rodríguez, and P.Sanz; EL1 | Escaped plant. | LEB 125014 |
| Villanueva de las Manzanas | León | 42.4771 −5.4879 | 23/08/21 | E.Alfaro-Saiz, S.Rodríguez, and P.Sanz; EL2 | Escaped plant. | LEB 125013 |
| Villanueva de las Manzanas | León | 42.4771 −5.4879 | 23/08/21 | E.Alfaro-Saiz, S.Rodríguez, and P.Sanz; EL3 | Escaped plant. | LEB 125012 |
| Villanueva de las Manzanas | León | 42.4771 −5.4879 | 23/08/21 | E.Alfaro-Saiz, S.Rodríguez, and P.Sanz; EL4 | Escaped plant. | LEB 125011 |
| Villanueva de las Manzanas | León | _ | 23/08/21 | E.Alfaro-Saiz, S.Rodríguez, and P.Sanz; EL5 | Cultivated. Magnum variety. Second year. Producer: Lúpulos Cantaleón. | |
| La Milla del Río | León | 42.57771949 −5.84540885 | 21/08/21 | D.López López and L.Martínez-Nieto; EL6 | Cultivated. Nugget variety. | LEB 125015 |
| La Milla del Río | León | 42.57242508 −5.85603174 | 21/08/21 | D.López López and L.M.Nieto; EL7 | Cultivated. Columbus variety. Twenty-one years. It has a strong smell. | LEB 125016 |
| La Milla del Río | León | 42.57345566 −5.85548222 | 21/08/21 | D.López López and L.M.Nieto; EL8 | Cultivated. Nugget variety. Twenty-one years. | LEB 125017 |
| La Milla del Río | León | 42.57771949 −5.84540885 | 21/08/21 | D.López López and L.M.Nieto; EL9 | Cultivated. Twenty years. | LEB 125018 |
| Oncina de la Valdoncina | León | 42.5527 −5.6597 | 05/09/21 | E.Alfaro-Saiz, A.B.Fernández-Salegui, J.M.Fos, and A.Fernández-Salegui; EL10 | Escaped plant. | LEB 125019 |
| Villoria de Órbigo | León | | 01/09/21 | S.Cámara-Leret, R. Martínez del Pozo D. López López, H. Buron de Lera, and S.Rodríguez; EL11 | Cultivated. Ecological crop. Chinook variety. Two years. Producer: Manolo. | LEB 125021 |
| Villamor de Órbigo | León | 42.481758 −5.870331 | 01/09/21 | S.Cámara-Leret, R.Martínez del Pozo, D.López López, H.Buron de Lera, and S.Rodríguez; EL12 | Cultivated. Nugget variety. Producer: J. C. Domínguez Matilla. | LEB 125020 |
| Villoria de Órbigo | León | | 01/09/21 | S.Cámara-Leret and R.Martínez del Pozo; EL13 | Cultivated. Millenium variety. More than 6 years. Producer: M. González Martínez. | LEB 125022 |

| Location | Province | Coordinates | Date (d/m/y) | Collectors | Observations | Voucher Number |
|---|---|---|---|---|---|---|
| San Roman de la Vega | León | 42.46877 −6.030442 | 25/08/21 | S.Cámara-Leret and R.Martínez del Pozo; EL14 | Cultivated. Eureka variety. One year. Producer: Alberto Martínez Ferrero (Hijo de "Talo"). | LEB 124210 |
| San Roman de la Vega | León | 42.4676606 −6.018388 | 25/08/21 | S.Cámara-Leret and R.Martínez del Pozo; EL15 | Cultivated. Cashmere variety. One year. Producer: Sergio Sánchez García. | LEB 124994 |
| Benavides de Órbigo | León | | 27/08/21 | S.Cámara-Leret; EL16 | Cultivated. Fifty years. Producer: Javier Rubio Mostrikles. | LEB 125028 |
| Gavilanes de Órbigo | León | 42.5207 −5.8835 | 24/08/21 | E.Alfaro-Saiz and R.Martínez del Pozo; EL17 | Cultivated. Forty years. Producer: Isidoro Alonso. | LEB 124998 |
| San Román de la Vega | León | 42.468010 −6.030363 | 25/08/21 | S.Cámara-Leret and R.Matínez del Pozo; EL20 | Cultivated. Nugget variety. Twenty-five years. Producer: Alberto Martínez Ferrero. | LEB 125000 |
| Benavides de Órbigo | León | 42.5090780 −5.8975700 | 24/08/21 | S.Cámara-Leret and H.Burón de Lera; EL21 | Cultivated. Producer: Javier "Los Triquis". | LEB 124996 |
| San Roman de la Vega | León | 42.468876 −6.020112 | 25/08/21 | S.Cámara-Leret and R.Martínez del Pozo; EL22 | Cultivated. Ecological crop. Four años. Producer: Sergio Sánchez García. | LEB 124997 |
| San Roman de la Vega | León | 42.468010 −6.030363 | 25/08/21 | S.Cámara-Leret and R.Martínez del Pozo; EL23 | Cultivated. Twenty-five years. Producer: Alberto Martínez Ferrero. | |
| Cuenca. El Chantre | Cuenca | | 07/09/21 | E.Saiz Flores; EL24 | Escaped plant. Male. | LEB 125006 |
| Cuenca. El Tranche | Cuenca | | 04/09/21 | E.Saiz Flores; EL 25 | Escaped plant. Female. | LEB 124971 |
| Nistal de la Vega | León | | 10/09/21 | D.López-López and M.Martínez-Vega; EL26 | Cultivated. Eureka variety. First year. | LEB 125003 |
| Nistal de la Vega | León | | 10/09/21 | D.López-López and M.Martínez-Vega; EL27 | Cultivated. Nugget variety. Six years. | LEB 125023 |
| Villaquilambre | León | 42.627533 −5.553374 | 13/09/21 | D.López-López; EL28 | | LEB 125002 |
| Ponferrada | León | 42.539 N 6.595 W | 25/10/21 | Ángel Argüelles; EL30 | Escaped plant. Male. Near a cultivated crop. | LEB 125008 |

| Location | Province | Coordinates | Date (d/m/y) | Collectors | Observations | Voucher Number |
|---|---|---|---|---|---|---|
| Cacabelos | León | 29T 686425 4719236 | 07/10/21 | E.Alfaro-Saiz, D.López-López, S.Rodríguez-Fernández, and L.González-Udaondo; EL31 | | LEB 125007 |
| Cuenca. El Chantre | Cuenca | 40°08′33.6″ N 2°08′12.2″ W | 12/08/21 | Carles Jiménez Box; EL32 | | LEB 125009 |
| Carrizo de la Ribera | León | 42°35′0.16″ N 5°49′17.94″ W | 08/09/21 | D.López-López; EL39 | Escaped plant. | LEB 124999 |
| Carrizo de la Ribera, en casa particular | León | 42.58834 −5.82538 | 12/09/21 | D.López-López; EL 40 | | LEB 125001 |
| Carrizo de la Ribera. Borde de Camino | León | 42°35′24.07″ 5°49′48.42″ | 04/08/21 | D.López-López; EL 41 | Escaped plant. | LEB 124972 |
| Carrizo de la Ribera. Borde de camino | León | 42.59139 −5.83654 | 04/08/21 | D.López-López; EL 42 | Escaped plant. Female. | LEB 125005 |
| Carrizo de la Ribera. Borde de camino | León | 42.60099 −5.83170 | 04/08/21 | D.López-López; EL 43 | | LEB 124973 |
| Turcia. Gavilanes de Órbigo | León | 42.5120 −5.8750 | 24/08/21 | E.Alfaro-Saiz, S.Cámara-Leret, R. Martínez del Pozo, and H. Burón de Lera; EL45 | Cultivated. Chinook variety. More than 20 years. Producer: Isidoro Alonso, "Los soles". | LEB 124857 |
| Carrizo de la Ribera | León | 42.60109477 −5.8316461 | 04/08/21 | D.López-López; EL 46 | Escaped plant. Male. | LEB 124858 |
| Gavilanes de Órbigo | León | | 24/08/21 | E.Alfaro-Saiz, S.Cámara-Leret, R.Martínez del Pozo, and H.Burón de Lera; EL47 | Escaped plant. Female. People from the village say that it is a wild plant. It is thought to have been there for many years. | LEB 124909 |
| Gavilanes de Órbigo | León | 42.5120 −5.8750 | 24/08/21 | E.Alfaro-Saiz, S.Cámara-Leret, R.Martínez del Pozo, and H.Burón de Lera; EL48 | Cultivated. Variety belonging to Hopsteiner. Three years. Producer: Isidoro Alonso ("Los soles"). | LEB 124952 |
| Gavilanes de Órbigo | León | 42.5120 −5.8750 | 24/08/21 | S.Cámara-Leret, E.Alfaro-Saiz, R.Martínez del Pozo, and Haide Burón de Lera; EL49 | Cultivated. More than 26 years. Producer: Isidoro Alonso ("Los soles"). | LEB 124953 |
| Gavilanes de Órbigo | León | 42.5120 −5.8750 | 24/08/21 | E.Alfaro-Saiz, S.Cámara-Leret, R.Martínez del Pozo, and H.Burón de Lera; EL50 | Cultivated. More than 24 years. Producer: Isidoro Alonso ("Los soles"). | LEB 124960 |

| Location | Province | Coordinates | Date (d/m/y) | Collectors | Observations | Voucher Number |
|---|---|---|---|---|---|---|
| Villoria de Órbigo | León | 42.40715722 −5.88247396 | 01/09/21 | S.Cámara-Leret, R.Martínez del Pozo, D.López López, H.Buron de Lera, and S.Rodríguez; EL51 | Cultivated. Ecological crop. Aquila variety. Producer: Manolo González Martínez. | LEB 124961 |
| Villoria de Órbigo | León | 42.40715722 −5.88247396 | 01/09/21 | S.Cámara-Leret, R.Martínez del Pozo, D.López López, H.Buron de Lera, and S.Rodríguez-Fernández; EL52 | Escaped plant. Growing near a fence next to Manolo González Martínez's shed. | LEB 124962 |
| Villoria de Órbigo | León | 42.39890552 −5.88142555 | 01/09/21 | S.Cámara-Leret, R.Martínez del Pozo, D.López López, H.Buron de Lera, and S.Rodríguez-Fernández; EL53 | Cultivated. Pacific Gem variety. One year. Producer: M.González Martínez. | LEB 124963 |
| Turcia. Gavilanes | León | 42.5120 −5.8750 | 24/08/21 | E.Alfaro-Saiz and R. Martínez del Pozo; EL54 | Cultivated. Alsacia variety.There was only one female plant on the seventh row. Sample sent to the laboratory. Producer: Isidoro Alonso. | LEB 124964 |
| Villaescusa de las Torres. | Palencia | 42.7597647 −4.2551251 | 16/08/21 | S.Rodríguez-Fernández; EL55 | Escaped plant. Female. | LEB 124965 |
| San Roman de la Vega | León | 42.467278 −6.017776 | 25/08/21 | S.Cámara-Leret and R.Martínez del Pozo; EL56 | Cultivated. Nugget variety. More than 20 years. Producer: Sergio Sánchez García. | LEB 124966 |
| Carrizo de la Ribera. | León | 42.58567704 −5.85114207 | 22/08/21 | D.López-López and A.García-Díez; EL57 | Cultivated. Nugget. | LEB 124967 |
| Quintanilla de Sollamas. | León | 42.60040795 −5.83329767 | 22/08/21 | D.López-López and A.García-Díez; EL58 | Cultivated. Nugget. | LEB 124968 |
| Nistal de la Vega | León | | 10/09/21 | D.López-López and M.Martínez-Vega; EL59 | Cultivated. Eureka variety. Second year. | LEB 125027 |
| Turcia. Gavilanes | León | 42.5233 −5.8713 | 24/08/21 | E.Alfaro-Saiz, S.Cámara-Leret, R.Martínez del Pozo, and H.Burón de Lera; EL60 | Cultivated. Five years. Producer: Isidoro Alonso. "Los Soles". | LEB 125029 |
| Cerezales del Condado | León | 42.710302 −5.362371 | 09/09/21 | D.López-López and L.Zurdo; EL61 | Escaped plant. | LEB 125030 |
| Vegas del Condado | León | 42.710289 −5.370203 | 09/09/21 | D.López-López and L.Zurdo; EL62 | Ambasaguas. | LEB 125031 |
| Vegas del Condado | León | 42.714778 −5.367711 | 09/09/21 | D.López-López and L.Zurdo; EL63 | Ambasaguas. | LEB 125032 |
| Carrizo de la Ribera | León | 42.587158 −5.823011 | 07/09/21 | D.López-López; EL64 | This plant probably belongs to the Alsacia variety. | LEB 125033 |

| Location | Province | Coordinates | Date (d/m/y) | Collectors | Observations | Voucher Number |
|---|---|---|---|---|---|---|
| Carrizo de la Ribera | León | 42.587348 −5.822861 | 07/09/21 | D.López-López; EL65 | This plant probably belongs to the Alsacia variety. | LEB 125034 |
| Calahorra de Ribas | Palencia | 42.164657 −4.533748 | 18/09/21 | S.Rodríguez-Fernández, and M.I.Cruz-Font; EL66 | Escaped plant. Female. | LEB 125035 |
| Calahorra de Ribas | Palencia | 42.164657 −4.533748 | 18/09/21 | S.Rodríguez-Fernández, and M.I.Cruz-Font; EL67 | Escaped plant. Male. | LEB 125036 |
| Villanueva de las Manzanas | León | 42.473253 −5.494907 | 02/09/22 | S.Cámara Leret and S.Rodríguez-Fernández; EL68 | Cultivated. Ecological crop. Producer: Gonzalo Pastrana. Lúpulos Cantaleón. There were no big leaves. Dry plants. The producer uses coconut fiber rope that sways in the wind. | LEB 125408 |
| Villamondrín | León | 42.573627 −5.283016 | 02/09/22 | S.Cámara Leret and S.Rodríguez-Fernández; EL69 | Escaped plant with no flowers. The plant was growing on a poplar at a riverside. | LEB 125409 |
| La Bañeza | León | 42.299507 −5.886030 | 05/09/22 | S.Cámara Leret, S.Rodríguez-Fernández, and D.López-López; EL70 | Escaped plant. Male. | LEB 125410 |
| La Bañeza | León | 42.299507 −5.886030 | 05/09/22 | S.Cámara Leret, S.Rodríguez-Fernández, and D.López-López; EL71 | Escaped plant. Female. | LEB 125411 |
| Villoria | León | | 05/09/22 | S.Cámara Leret, S.Rodríguez-Fernández, and D.López-López; EL72 | Cultivated. H3 variety. Seventy years. Producer: Manolo González Martínez. La cultivaba su abuelo. | LEB 125412 |
| Villoria | León | | 05/09/22 | S.Cámara Leret, S.Rodríguez-Fernández, and D.López-López; EL73 | Cultivated. Eureka variety. Two years. Producers do not like this variety because it has a strong smell. The smell makes you cough. Producer: Pedro Cuevas. | LEB 125413 |
| Villoria | León | | 05/09/22 | S.Cámara Leret, S.Rodríguez-Fernández, and D.López-López; EL74 | Cultivated. H7 variety. Seventy years. Producer: Manolo González Martínez. His grandfather cultivated this variety. | LEB 125414 |

| Location | Province | Coordinates | Date (d/m/y) | Collectors | Observations | Voucher Number |
|---|---|---|---|---|---|---|
| San Román de los Caballeros | León | 42.655163 −5.837086 | 05/09/22 | S.Cámara Leret, S.Rodríguez-Fernández, and D.López-López; EL75 | Cultivated. Willamette variety. Aromática. Four years. Producer: Enrique Sevilla García. | LEB 125415 |
| San Román de los Caballeros | León | 42.657772 −5.846190 | 05/09/22 | S.Cámara Leret, S.Rodríguez-Fernández, and D.López-López; EL76 | Cultivated. Admiral variety. One year. Producer: Enrique Sevilla García. | LEB 125416 |
| San Román de los Caballeros | León | 42.657772 −5.846190 | 05/09/22 | S.Cámara Leret, S.Rodríguez-Fernández, and D.López-López; EL77 | Cultivated. Chinook variety. Tallo grueso. Aromática. Producer: Enrique Guilla García. | LEB 125417 |
| San Martín de la Falamosa | León | 42.693589 −5.881472 | 05/09/22 | S.Cámara Leret, S.Rodríguez-Fernández, and D.López-López; EL78 | Escaped plant. | LEB 125418 |
| San Martín de la Falamosa | León | 42.701107 −5.900519 | 05/09/22 | S.Cámara Leret, S.Rodríguez-Fernández, and D.López-López; EL79 | Escaped plant. | LEB 125419 |
| San Feliz de Torío | León | 42.6671684 −5.539581 | 06/09/22 | S.Rodríguez-Fernández, D.López-López, and E.Diez-Presa; EL80 | Escaped plant. | LEB 125420 |
| Garrafe de Torío | León | 42.7327061 −5.5217864 | 06/09/22 | S.Rodríguez-Fernández, D.López-López, and E.Diez-Presa; EL81 | Escaped plant. | LEB 125422 |
| San Justo de la Vega | León | 42.4810053 −6.037692 | 07/09/22 | S.Cámara Leret, S.Rodríguez-Fernández, and D.López-López; EL82 | Cultivated. Sorachi variety. One year. It scratches a lot. Citrus smell. Producer: Sergio and Diego Sánchez. | LEB 125423 |
| San Román de la Vega | León | 42.46884554 −6.035260 | 07/09/22 | S.Cámara Leret, S.Rodríguez-Fernández, and D.López-López; EL83 | Escaped plant. Growing on a burned tree. | LEB 125424 |
| San Justo de la Vega | León | 42.467235 −6.018990 | 07/09/22 | S.Cámara Leret, S.Rodríguez-Fernández, and D.López-López; EL84 | Cultivated. Cashemere variety. Two years. Producer: Sergio y Diego Sánchez | LEB 125425 |
| San Justo de la Vega | León | 42.467235 −6.018990 | 07/09/22 | S.Cámara Leret, S.Rodríguez-Fernández, and D.López-López; EL85 | Cultivated. Cascade variety. Producer: Sergio and Diego Sánchez | LEB 125426 |
| Sahagún | León | 42.375027 −5.034118 | 10/09/22 | D.López-López; EL86 | It has a strong bitter smell. It scratched a lot. | LEB 125427 |

| Location | Province | Coordinates | Date (d/m/y) | Collectors | Observations | Voucher Number |
|---|---|---|---|---|---|---|
| Carrizo de la Ribera | León | 42.595047 −5.828480 | 09/09/22 | S.Cámara Leret, D.López-López, and S.Rodríguez-Fernández; EL87 | Cultivated. Magnum variety. Thirty years. Producer: Asunción Martínez Rodríguez. It has more inflorescences than other varieties and most of them are bigger. | LEB 125445 |
| León | León | 42.59640 −5.548712 | 06/09/22 | S.Rodríguez-Fernández, and E.Diez-Presa; EL88 | | LEB 125446 |
| Villaobispo | León | 42.6275 −5.553456 | 25/07/22 | D.López-López; EL89 | | LEB 125447 |
| Carrizo de la Ribera | León | 42.588825 −5.841734 | 09/09/22 | S.Cámara Leret, D.López-López, and S.Rodríguez-Fernández; EL90 | Cultivated. Nugget variety. Twenty-five years. Producer: Ángel. | LEB 125448 |
| Carrizo de la Ribera | León | 42.588329 −5.848299 | 09/09/22 | S.Cámara Leret, D.López-López, and S.Rodríguez-Fernández; EL91 | Cultivated. Twenty years. Producer: Fernando Fernández Magaz. | LEB 125449 |
| Carrizo de la Ribera | León | 42.595047 −5.828480 | 09/09/22 | S.Cámara Leret, D.López-López, and S.Rodríguez-Fernández; EL92 | Cultivated. Nugget variety. More than 30 years. Producer: Asunción Martínez Rodríguez. | LEB 125450 |
| Villaverde de Omaña | León | 42.782737 −6.094837 | 28/08/22 | D.López-López; EL93 | People from the village say that the plant was planted there ten years ago. | LEB 125451 |
| Cirujales. | León | 42.784325 −6.077643 | 28/08/22 | D.López-López; EL94 | People from the village say that the plant was planted there a couple of years ago. | LEB 125452 |
| Villaverde de Omaña | León | | | E.Alfaro-Saiz, S.Cámara Leret, S.Rodríguez-Fernández, D.López-López, E.Diez, M.Martínez, and A.Valiente; EL95 | | LEB 125453 |
| León | León | 42.590401 −5.575946 | 09/07/22 | S.Rodríguez-Fernández; EL96 | | LEB 125454 |
| Villaobispo | León | 42.627580 −5.553456 | 25/07/22 | D.López-López; EL97 | | LEB 125455 |
| Cerezales del Condado | León | 42.717405 −5.359431 | 17/08/22 | E.Alfaro-Saiz, S.Cámara Leret, A.Andrés, S.Rodríguez, D.López, E.Diez, M.Martínez, and A.Valiente; EL98 | | LEB 125456 |

| Location | Province | Coordinates | Date (d/m/y) | Collectors | Observations | Voucher Number |
|---|---|---|---|---|---|---|
| Ponferrada | León | 42.8708322 −5.4450192 | 12/09/22 | E.Diez-Presa; EL99 | | LEB 125459 |
| Villa del Prado | Madrid | 40.237972 −4.273454 | 30/09/22 | C.Parro-Arellano; EL100 | There is no evidence of cultivation in this area. In the spring, people consume the shoots called "zaramangones". | LEB 125460 |
| Montardit de Baix | Lérida | 42°22′20″ N 4°06′38″ E | 20/08/22 | C.Burguera Martín; EL101 | | LEB 125461 |
| Abia de las Torres | Palencia | 42.4230760 −4.4184730 | 11/09/22 | S.Diez-Camino; EL102 | | LEB 125463 |

**Appendix B**

Biological activity of the selected compounds present in the essential oil from *Humulus lupulus*.

| Chemical Group | Chemical Class | Phytochemical | Biological Activity | References |
|---|---|---|---|---|
| Alcohol | Fatty alcohol | 2-Methyl-3-buten-2-ol | Sedative Narcotic Somniferous Liver protector | - McGinty et al., 2010 [84]<br>- Franco et al., 2012 [85]<br>- Schepetkin et al., 2022 [86]<br>- Gray, 2002 [87]<br>- Mannering and Shoeman, 1996 [88] |
| Sesquiterpenes | Monocycles sesquiterpenes | α-caryophyllene | Anti-inflammatory Analgesic Anesthetic Antitumoral Antioxidant Immunomodulator Neuroprotective Antidiabetic Antihistamine Antimicrobial Narcotic Antidepressant Anxiety Delirium Anxiolytic | - Viveiros et al., 2022 [61]<br>- Gullì et al., 2022 [54]<br>- Rossato et al., 2022 [89]<br>- Nuutinen, 2018 [57]<br>- Santos et al., 2022 [59] |
| | | Germacrene-D | Antimicrobial Anti-inflammatory Anticarcinogen Antioxidant Immunomodulator Spasmolytic Antihistamine | - Azimi et al., 2011 [90]<br>- Santos et al., 2022 [59]<br>- Saab et al., 2018 [58]<br>- Mukarram et al., 2021 [91] |

| Chemical Group | Chemical Class | Phytochemical | Biological Activity | References |
|---|---|---|---|---|
| Sesquiterpenes | Bicycles sesquiterpenes | β-caryophyllene | Anti-inflammatory Analgesic Anesthetic Antitumoral Antioxidant Antidiabetic Neuroprotector Antimicrobial Immunomodulator Spasmolytic Antihistamine | - Viveiros et al., 2022 [61]<br>- Gullì et al., 2022 [54]<br>- Hu et al., 2017 [55]<br>- Scandiffio et al., 2020 [60]<br>- Santos et al., 2022 [59]<br>- Saab et al., 2018 [58]<br>- Nuutinen, 2018 [57] |
| | Acyclic sesquiterpenes | β-farnesene | Anticarcinogen Antimicrobial Antioxidant Neuroprotective | - Turkez et al., 2014 [92] |
| Sesquiterpenoids | Tertiary alcohol | Humulol | Antitumoral Anti-inflammatory | - Yasukawa et al., 1995 [93] |
| Flavonoid | Flavonones | 8-Prenylnaringenin | Anticarcinogen Estrogenic activity Anti-inflammatory Antioxidant Osteoprotective Cardioprotective | - Arranz et al., 2012 [53]<br>- Estruch et al., 2010 [52] |
| | Flavonols | Kaempferol | Antioxidant Anticarcinogen Anti-inflammatory Antimicrobial Antiadipogenic Spasmolytic Antidiabetic Cardioprotective Osteoprotective Relaxing | - Periferakis et al., 2022 [94]<br>- Las Heras Etayo et al., 2021 [56]<br>- Chen et al., 2014 [72] |
| | Prenylflavonoids | Isoxanthohumol | Anticarcinogen Anti-inflammatory Antioxidant Osteoprotective Antimicrobial Estrogenic activity | - Milligan et al., 2000 [62]<br>- Arranz et al., 2012 [53]<br>- Estruch et al., 2010 [52]<br>- Chen et al., 2014 [72] |
| Terpenes | Bicyclic monoterpenes | α-pinene | Antitumoral Anticarcinogen Antioxidant Anti-inflammatory Analgesic Antihistamine Anxiolytic Hypnotic Bronchodilator Antimicrobial | - Nuutinen, 2018 [57]<br>- Santos et al., 2022 [59] |

| Chemical Group | Chemical Class | Phytochemical | Biological Activity | References |
|---|---|---|---|---|
| Terpenes | Bicyclic monoterpenes | β-pinene | Antidepressant<br>Sedative<br>Antinociceptive<br>Antihypertensive<br>Endotheliogenesis<br>Antiviral<br>Antiadipogenic<br>Antidiabetic<br>Antimicrobial<br>Contraceptive | - Nuutinen, 2018 [57]<br>- Santos et al., 2022 [59] |
| | Acyclic monoterpene | Myrcene | Sedative<br>Somniferous<br>Anti-inflammatory<br>Osteoprotective<br>Cardioprotective<br>Antioxidant<br>Neuroprotective<br>Hepatogenesis<br>Antiulcer<br>Contraceptive<br>Antimutagenic<br>Anticancer | - Nuutinen, 2018 [57] |
| | | Geraniol | Anticarcinogen<br>Anti-inflammatory<br>Antidiabetic<br>Cardioprotective<br>Antiadipogenic<br>Vasoprotector<br>Antioxidant<br>Inmunoestimulador<br>Hepatogenesis<br>Lung protector<br>Antidepressant<br>Neuroprotective<br>Antimicrobial | - Nuutinen, 2018 [57]<br>- Azimi et al., 2011 [90]<br>- Mukarram et al., 2021 [91] |
| | | Linalool | Anxiolytic<br>Sedative<br>Analgesic<br>Anticonvulsant<br>Antitumoral<br>Anti-inflammatory<br>Antimicrobial<br>Neuroprotective<br>Anticarcinogen<br>Antidepressant<br>Antihypertensive<br>Antioxidant<br>Hepatoprotective<br>Antimicrobial | - Nuutinen, 2018 [57]<br>- Azimi et al., 2011 [90] |

| Chemical Group | Chemical Class | Phytochemical | Biological Activity | References |
|---|---|---|---|---|
| Terpenes | Monoterpenes cyclohexanes | Limonene | Anxiolytic Inmunoestimulator Anticancer Chemopreventive Antitumoral Anti-inflammatory Antihyperalgesics Analgesic Osteoprotective Antioxidant Sedative Relaxing Antidepressant Antispasmodic Antiviral Healing | - Nuutinen, 2018 [57]<br>- Azimi et al., 2011 [90] |
| Ketones | Aliphatic ketone | 2-Undecanone | Anti-inflammatory Antioxidant | - Wu et al., 2021 [95]<br>- Jiao et al., 2022 [96] |
| | Cyclohexanone | Cohumulone | Antitumoral Anticarcinogen Antibacterial | - Estruch et al., 2010 [52]<br>- Lewis et al., 1949 [97]<br>- Bogdanova et al., 2018 [98]<br>- Arranz et al., 2012 [53] |
| | | Ad humulone | Antimicrobial Antitumoral | - Li et al., 2023 [99]<br>- Arranz et al., 2012 [53] |
| | | Colupulone | Antimicrobial | - Li et al., 2023 [99] |
| | | Ad lupulone | Antimicrobial | - Li et al., 2023 [99] |
| | | Lupulone | Antimicrobial Anticancer | - Milligan et al., 2000 [62]<br>- Arranz et al., 2012 [53]<br>- Estruch et al., 2010 [52]<br>- Chen et al., 2014 [72] |
| | Chalcone | Xanthohumol | Anticarcinogen Osteoprotective Anti-inflammatory Antioxidant Antitumoral Antimicrobial | - Arranz et al., 2012 [53]<br>- Estruch et al., 2010 [52]<br>- Bogdanova et al., 2018 [98]<br>- Chen et al., 2014 [72] |

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
