# Peer review of "The Memory of Hops: Rural Bioculture as a Collective Means of Reimagining the Future"

_sustainability, doi:10.3390/su16062470_

Round 1

Reviewer 1 Report

Comments and Suggestions for Authors After analysis of the published manuscript: The memory of hops: rural bioculture as a collective means of reimagining the future, by Estrella Alfaro-Saiz , Susana Câmara-Leret, Miguel González-Gonzále, Óscar Fernández-Álvarez, Sergio Rodríguez-Fernández, Darío López-López, Ana Isabel Paniagua-García, Carmem Acedo, Rebeca Díez-Antolínez , I make some observations. I was surprised to see 9 authors on this manuscript – author inflation is a concern for all journals today, it would be important to be cautious even if there is no restriction by the journal on a maximum or minimum number. The manuscript deals with the cultivation of Humulus lupulus in Spain (2nd largest in Europe), with the proposal basically around 4 aspects: a herbarium and living bank of the species found (along with a map), a chemical analysis of the compounds that may have pharmacological action, local interview with some producers to understand the problems and dynamics and guidance for them to develop new products from Humulus lupulus leftovers. Overall the text is good, but I noticed that sometimes the authors deviate from the topic. There are times when the discussion becomes a true bibliographical review instead of discussing what they found, and at the same time, I don't think they explored their data so well, it's just text and, in this manuscript, graphs can be presented to facilitate understanding of the general situation of producers. In this sense, was there authorization from the research ethics committee to use participant information? Which communities did they interview? Essential information is missing; an interview was carried out, but the questions and any type of classification were not presented, not even in supplementary material. I think as a general guideline I would tell them to reduce the text, be more direct, focus on observations based on the research results and review the English!

Introduction. It’s well written. However, I believe the Introduction is too long, it’s easy to get lost in the middle of so much information and lose sight of what’s essential to better comprehend this work. Maybe, create a separated section further to clarify the historic of the region would help to shorten the text.

Ethnographic Interviews. Even if you mentioned the reference, the way how you conducted the interviews, the questions, the objectives and how it was oriented remains a mystery. The authors should add at least a table with the questions asked, and maybe the areas you wanted to figure out.

Ethnographic Interviews.Why not make graphs to better illustrate themes of questions, the genre of answers and all the patterns you figure out? In my opinion this topic is not well explored.

item 3.4 a. the ratio amount of plant need/ amount of fibre would be utile

item 3.4 b. Did test pigmentation, any materials it would be applicable to?

item 3.4 c. the ratio amount of plant need/ amount paper produced would be very utile

Discussion. the authors didn’t really cite the herbarium species you collected during the study. Didn’t you get any patterns, conclusions or important information from?

Biocompounds & Cultivars. In general, the possible use of those compounds in healthcare are already known and studied. My is suggestion is, rather than focus your discussion detailed explaining these actions, try to question why these plants are not well explored. If the pharmacy industry has limitations to it, would it be possible to make some sort of traditional medication or tea in order to help with their economic limitations?

Bioproducts vs. Waste. An idea: Wouldn’t these products demand less physical work? If that is true, what are the perspectives of using these products to help keeping younger people in the field? As you show in your Youtube videos, the younger generation tend to don’t appreciate labor work.

Comments on the Quality of English Language

Minor editing of English language required

Author Response

A: I greatly appreciate your effort in reviewing our manuscript. Below, you will find detailed responses, with the respective revisions and corrections clearly marked or tracked in the files we are resubmiting. As Reviewer 1 suggests, we incorporated a table of the topics raised in the interviews, rather than questions, since we did not conduct surveys. Likewise, it is correlated with the reporting actors and the scope of the research criterion. We appreciate this suggestion, which results in greater systemic clarity and research methodology.

R1: After analysis of the published manuscript: The memory of hops: rural bioculture as a collective means of reimagining the future, by Estrella Alfaro-Saiz , Susana Câmara-Leret, Miguel González-Gonzále, Óscar Fernández-Álvarez, Sergio Rodríguez-Fernández, Darío López-López, Ana Isabel Paniagua-García, Carmem Acedo, Rebeca Díez-Antolínez, I make some observations. I was surprised to see 9 authors on this manuscript – author inflation is a concern for all journals today, it would be important to be cautious even if there is no restriction by the journal on a maximum or minimum number. 

A: In response to concerns about the number of authors, each author has significantly contributed to this multidisciplinary project and research , which integrates various perspectives and methodologies (botany, art, anthropology, etc.). Additional contributors, not listed as authors, have also played essential roles, highlighting the collaborative essence of this research. To exclude contributors due to numerical limits would be more problematic. Recognising every significant contributor is vital in multidisciplinary research, where diverse expertise is crucial. We aim to maintain a balance that honors the collective scientific work, ensuring fairness and transparency in academic publishing.

R1: The manuscript deals with the cultivation of Humulus lupulus in Spain (2nd largest in Europe), with the proposal basically around 4 aspects: a herbarium and living bank of the species found (along with a map), a chemical analysis of the compounds that may have pharmacological action, local interview with some producers to understand the problems and dynamics and guidance for them to develop new products from Humulus lupulus leftovers. 

R1: Overall the text is good, but I noticed that sometimes the authors deviate from the topic. There are times when the discussion becomes a true bibliographical review instead of discussing what they found, and at the same time, I don't think they explored their data so well, it's just text and, in this manuscript, graphs can be presented to facilitate understanding of the general situation of producers.

R1: In this sense, was there authorization from the research ethics committee to use participant information? 

A: We confirm that all people interviewed for this study provided their informed consent for the use of data generated during the interviews. Prior to their participation, each interviewee was fully informed about the study's objectives, the nature of their contribution, and the intended use of the information collected. Regarding informed consent, it was mentioned in the manuscript’s text, more specifically in lines 268-269. Nevertheless, we highlight the following paragraph in the text: There was an informed consent, all people agreed to participate voluntarily in the research and knew the criteria and purposes of the project.

R1: Which communities did they interview? Essential information is missing; an interview was carried out, but the questions and any type of classification were not presented, not even in supplementary material.

A: Done. Regarding the communities interviewed, please view table 3 

R1: I think as a general guideline I would tell them to reduce the text, be more direct, focus on observations based on the research results and review the English!

R1: Introduction. It’s well written. However, I believe the Introduction is too long, it’s easy to get lost in the middle of so much information and lose sight of what’s essential to better comprehend this work. Maybe, create a separated section further to clarify the historic of the region would help to shorten the text.

A: We have taken your advice into account and condensed the Introduction to focus more succinctly on the essential aspects of our work. This adjustment should make it easier for readers to grasp the fundamental concepts and significance of our research.

R1: Ethnographic Interviews. Even if you mentioned the reference, the way how you conducted the interviews, the questions, the objectives and how it was oriented remains a mystery. The authors should add at least a table with the questions asked, and maybe the areas you wanted to figure out.

R1: Ethnographic Interviews.Why not make graphs to better illustrate themes of questions, the genre of answers and all the patterns you figure out? In my opinion this topic is not well explored.

A: Done, please see table 3

R1: item 3.4 a. the ratio amount of plant need/ amount of fibre would be utile

A: For the purpose of our study we focused on establishing a correspondence to similar cellulose fibre extraction processes, like linen and hemp, to confirm the obtainment of fibre from the plant’s stem. We hope to expand on this in the next stages of the project, as we have received additional support to develop further this line of work.  

R1: item 3.4 b. Did test pigmentation, any materials it would be applicable to?

A: Thank-you for this observation. As indicated in Materials & Methods, the dyeing tests were done on protein and cellulose-based fabrics. We have added the following sentence to the Results section: These colour tests are suitable for fabric dyeing.

R1: item 3.4 c. the ratio amount of plant need/ amount paper produced would be very utile

A: We appreciate your suggestion. As we used the paper pulp to create masks, tile samples and other objects (to understand the possibilities of the material) we did not quantify its output (plant matter per paper sheet). To provide a reference, we added the following sentence to the results:

A: The following paragraph was addedFrom 1 kg of plant matter two 300 x 300 x 30 mm tiles and two 200 x 100 x 70 mm bricks were produced with a considerable amount of leftover material for the creation of additional objects and paper sheets. 

R1: Discussion. the authors didn’t really cite the herbarium species you collected during the study. Didn’t you get any patterns, conclusions or important information from?

A: Done. The following paragraph was added to the discussion. Thank you very much for the suggestion:

A: The herbarium specimens collected during fieldwork are the vouchers of the localities we confirmed. In addition, these specimens allowed us to link the current presence of the species with the crop map published by Breuer in 1985 in the province of León [1]. The absence of samples in areas where crops have not existed leads us to consider the possibility that the plant is no longer in the wild. 

R1: Biocompounds & Cultivars. In general, the possible use of those compounds in healthcare are already known and studied. My is suggestion is, rather than focus your discussion detailed explaining these actions, try to question why these plants are not well explored. If the pharmacy industry has limitations to it, would it be possible to make some sort of traditional medication or tea in order to help with their economic limitations?

A: Done. The following paragraph was added to the discussion. Thank you very much for the suggestion:

A: Addressing the underuse of hop compounds in healthcare despite known benefits, this study highlights the pharmacological potential of Humulus lupulus, especially its bioactive compounds with therapeutic effects validated by clinical trials. Regulatory hurdles and the pharmaceutical industry's preference for synthetic drugs may restrict these compounds' use in mainstream healthcare. However, our findings recommend further exploration of hops as alternatives to harness these beneficial compounds.

R1: Bioproducts vs. Waste. An idea: Wouldn’t these products demand less physical work? If that is true, what are the perspectives of using these products to help keeping younger people in the field? As you show in your Youtube videos, the younger generation tend to don’t appreciate labor work.

A: Indeed as you suggest these products would demand less physical work. Our initial results are critical for, firstly, creating an awareness on the need for more detailed research on creating new biomaterials from hop waste. This will have to include a more detailed study on local waste management systems and services which on this occasion was beyond the scope of our study but which we nonetheless hope to contribute to in the next phases of our project and ongoing research. As we suggest in our study, the potential diversification of hop cultivation and applications would improve the precariousness of hop’s harvest (improving the crop’s prices) making it more appealing to the younger generations. Moreover, as we show in our work, inserting its byproducts into cultural networks (triggering other uses of its fibers, stems, etc) for example through local traditional practices such as the Antruejo, could trigger other alternative uses of the crop within local communities, changing how the younger generations relate to these products.  

Reviewer 2 Report

Comments and Suggestions for Authors

The submitted manuscript presents a multidisciplinary study that explored new relationships with hops and examined its ecological and ethnobotanical aspects, traditional uses, and potential applications beyond brewing, such as in pharmacy and cosmetics. The study utilized a qualitative ethnographic approach and art-based material experimentation to show the many possibilities of hops and the potential impacts on agricultural sustainability and territorial cohesion. Overall the paper is well-written, but some points should be better demonstrated.

1. Scope of the study. The study focused on a specific region and its traditional hop cultivation practice and therefore lacks the broader implications for agricultural sustainability in different contexts. The nature of the study (a case study basic in a specific geographical area) shall be noted in the title and the limitations regarding the findings' generalizability should be discussed. 

2. Sample selection in the interviews. Are the selected participants for the interviews adequately representative of the broader community involved in hop cultivation? If not, the investigated group (8 farmers and 4 experts) may introduce bias.

3. Material experiments. While innovative, the reproducibility and scalability of creating new biomaterials from hop waste may face challenges in practical application without further research. For example, how feasible is it to implement the waste processing methods on a larger scale? To be more rigorous, the environmental benefits of reducing waste should be further evaluated through life-cycle analysis.

Author Response

R2: The submitted manuscript presents a multidisciplinary study that explored new relationships with hops and examined its ecological and ethnobotanical aspects, traditional uses, and potential applications beyond brewing, such as in pharmacy and cosmetics. The study utilized a qualitative ethnographic approach and art-based material experimentation to show the many possibilities of hops and the potential impacts on agricultural sustainability and territorial cohesion. Overall the paper is well-written, but some points should be better demonstrated.

A: I greatly appreciate your effort in reviewing this manuscript and improving it. Below, you will find detailed responses to your suggestions, with the respective revisions and corrections marked or tracked in the files resubmitted.

R2: 1. Scope of the study. The study focused on a specific region and its traditional hop cultivation practice and therefore lacks the broader implications for agricultural sustainability in different contexts. The nature of the study (a case study basic in a specific geographical area) shall be noted in the title and the limitations regarding the findings' generalizability should be discussed. 

A: Thank you very much for the suggestion. We have revised the title to better reflect the study's specific focus and geographical context.

A: In addition, the following paragraph was added to the discussion: This study, focused on León's traditional hop cultivation, presents a detailed exploration of its unique contribution to agricultural sustainability within a specific locale. Aside from offering valuable localized insights, we recall that León produces 95% of all spanish hop production and it is the tenth european producer (see lines 61-64). Therefore, the study's geographic specificity is also relevant to the applicability of our findings to other regions. Future research should nonetheless aim to compare these practices across varied geographical contexts to extend the relevance of sustainable agricultural practices in other places.

R2: 2. Sample selection in the interviews. Are the selected participants for the interviews adequately representative of the broader community involved in hop cultivation? If not, the investigated group (8 farmers and 4 experts) may introduce bias.

A: In response to your question regarding sample selection for interviews, accessibility to the broader community poses challenges; however, the participants selected (see table 3) provide a meaningful representation, especially considering their active involvement and expertise in hop cultivation. The group of 8 farmers and 4 experts, while not encompassing the entire community, offers valuable insights from those willing to collaborate. This approach ensures that our study captures critical perspectives, albeit from a subset of the community eager to share their experiences.

A: The following paragraph was added.

A: In addition, open-ended group interviews were conducted from October 2021 to August 2023, comprised six open group sessions with varying numbers of participants. The first session, "Voces de otro tiempo," was held in October 2021 in Gavilanes de Órbigo with 16 attendees. "Ir a lúpulo" followed in August 2022 at Fundación Cerezales Antonino y Cinia with 55 participants. "El lúpulo y su futuro" took place in November 2022 in Carrizo de la Ribera, engaging 45 participants. January 2023 saw two sessions in Carrizo de la Ribera: "Textile dyes" with 32 participants and "Basket weaving" with 40 attendees. The series concluded with "Entramados: Hops and its fibers" in August 2023 at the Ethnographic Museum of León with 24 participants. These sessions offered diverse insights into the cultural and practical aspects of hop cultivation and utilization.

R2: 3. Material experiments. While innovative, the reproducibility and scalability of creating new biomaterials from hop waste may face challenges in practical application without further research. For example, how feasible is it to implement the waste processing methods on a larger scale? To be more rigorous, the environmental benefits of reducing waste should be further evaluated through life-cycle analysis.

A: Thank you for your observation. This phase is indeed a preliminary exploration aimed at assessing the potential for utilizing hop waste in biomaterial production. Whilst we acknowledge the possible challenges in reproducibility and scalability, these initial results are critical for paving the way to more detailed research on creating new biomaterials from hop waste. This will have to include a more detailed study on local waste management systems and services (eg. drying processes, processing methods and overall output), which on this occasion  was beyond the scope of our study. We nonetheless hope to contribute to it in the next phases of our project and ongoing research.

Reviewer 3 Report

Comments and Suggestions for Authors

I like the article, although it also deeply bothered me at times. I still do not see the need to be grandiloquent when simple phrases will do. I invite authors to express their thoughts with the goal of being understood rather than followed (as adepts do).

It is a generally widespread notion, not necessarily accepted by all, that plants can cure almost any human ailment. There is no doubt that plants are nature's chemical masters and there is almost always a phytochemical capable of performing an activity that helps relieve pain, fight bacteria or other pathogens, and regulate certain metabolic activities. It is also true, however, that even if valid in terms of phytochemical effects, what science has discovered is that the concentration of most of those chemicals is so low that their true relevance as substitutes for other forms of therapies is questionable. In short, I recommend moderation when reaching grandiose conclusions.

The authors commendably presented the result of research involving many different approaches. What I could not see was a discussion that encompassed the more expected multidimensional discussion of the complete results. Each particular component was expressed in terms of its own, particular narrative– when authors invited us to think more globally instead. I recommend the authors present us with a much more coherent, articulate and sound discussion that serves the purpose of introducing the conclusions that will follow.

Conclusions were easy to understand and accept, except for the last paragraph, once the pamphletary nature of the Discussion was overcome.

The article has its value, but the descriptive job should be revised to present a more palatable, non-pontificating scientific article that I am sure will be of interest to many.

Comments on the Quality of English Language

I detected instances in which English was used with a kind of nostalgia for Spanish syntax. It did not bother me, but others might.

Author Response

R4:I like the article, although it also deeply bothered me at times. I still do not see the need to be grandiloquent when simple phrases will do. I invite authors to express their thoughts with the goal of being understood rather than followed (as adepts do).

R4: It is a generally widespread notion, not necessarily accepted by all, that plants can cure almost any human ailment. There is no doubt that plants are nature's chemical masters and there is almost always a phytochemical capable of performing an activity that helps relieve pain, fight bacteria or other pathogens, and regulate certain metabolic activities. It is also true, however, that even if valid in terms of phytochemical effects, what science has discovered is that the concentration of most of those chemicals is so low that their true relevance as substitutes for other forms of therapies is questionable. In short, I recommend moderation when reaching grandiose conclusions.

A: We emphasize the rigorous scientific methodology applied in our study. Utilizing the Swiss Target Prediction (STP) tool, we've conducted analyses on local plant biocompounds, ensuring a robust examination of their pharmacological potential. This approach allows us to validate the therapeutic effects of these compounds. Our findings contribute to a nuanced understanding of the medicinal value of plants, grounded in empirical evidence.

R4: The authors commendably presented the result of research involving many different approaches. What I could not see was a discussion that encompassed the more expected multidimensional discussion of the complete results. Each particular component was expressed in terms of its own, particular narrative– when authors invited us to think more globally instead. I recommend the authors present us with a much more coherent, articulate and sound discussion that serves the purpose of introducing the conclusions that will follow.

A: Thank you very much for the suggestion. The following paragraph was added to the discussion, emphasizing on the multidisciplinary approach and findings of our study: 

A: This approach bridges cultural heritage, sustainable agriculture, and innovative material uses, underscoring the interplay between traditional practices and modern sustainability challenges. By valuing diversity in all its forms—languages, economic activities, cultivation systems, and crop varieties—it fosters a holistic understanding of hop cultivation's impact on its territory. This strategy not only promotes resilience and sustainability in rural communities but also serves as a model for establishing territorial guidelines and pilot models across different geographical contexts. This convergence of disciplines offers a path towards a sustainable, culturally rich future, highlighting the transformative potential of traditional crops like hops in addressing contemporary challenges.

R4: Conclusions were easy to understand and accept, except for the last paragraph, once the pamphletary nature of the Discussion was overcome.

A: The last paragraph has been changed to improve the message, based on our findings, and to enhance an integrative view of the disciplines.

R4: The article has its value, but the descriptive job should be revised to present a more palatable, non-pontificating scientific article that I am sure will be of interest to many.

A: Thank you, we have revised the article to be more succinct.

Reviewer 4 Report

Comments and Suggestions for Authors

\this is a very long paper.   But had it be written in short, sharp sentences, omitting many subtleties, it would have lost its value.  It tell an extensive story about how hop cultivatioon is embedded in the community, how it is pressuered by outside forces, and how the result is de-population, the young emigrating leaving the old behind.  As such it is a story that is repeated in many part of the world;  this thorough study of hops is valuable as a well documented example.

As a scientific paper it could be much shortened;  but its value is its story and the telling of the complexities of the itnerplay of cultivation and community.   Thank you for an intersting paper

Author Response

R3: this is a very long paper.   But had it be written in short, sharp sentences, omitting many subtleties, it would have lost its value.  It tell an extensive story about how hop cultivatioon is embedded in the community, how it is pressuered by outside forces, and how the result is de-population, the young emigrating leaving the old behind.  As such it is a story that is repeated in many part of the world;  this thorough study of hops is valuable as a well documented example. As a scientific paper it could be much shortened;  but its value is its story and the telling of the complexities of the itnerplay of cultivation and community.   Thank you for an intersting paper

A: Thank-you for your encouraging recommendations.  

Round 2

Reviewer 1 Report

Comments and Suggestions for Authors In my observations I noticed that the authors attended all the suggestions indicated in the review and the text was very good.
The manuscript has potential for publication.

Author Response

Your insightful feedback has been pivotal in refining our work, ensuring clarity, and enhancing its contribution to the field.

Thank you,

Estrella

Reviewer 2 Report

Comments and Suggestions for Authors

Many thanks to the authors for addressing my concerns. The revisions made have significantly improved the manuscript, and I am pleased to accept the revised article for publication.

Author Response

Your insightful feedback has been pivotal in refining our work, ensuring clarity, and enhancing its contribution to the field.

Thank you 

Estrella